# Impact of maternal compensation on developmental phenotypes in a zebrafish model of severe congenital muscular dystrophy

Kyle P. Flannery[1], Shorbon Mowla[1], Namarata Battula[1], L. Rose Clark[1], Callista D. Oliveira[1], Lillian M. Simhon[1], Deze Liu[2], Cynthia Venkatesan[1], Brittany F. Karas[1], Kristin R. Terez[1], Daniel Burbano[2], M. Chiara Manzini[1]*

1 Department of Neuroscience and Cell Biology, Rutgers-Robert Wood Johnson Medical School, Child Health Institute of New Jersey, New Brunswick, New Jersey, United States of America, 2 Department of Electrical and Computer Engineering, Rutgers University, Piscataway, New Jersey, United States of America

* chiara.mazini@rutgers.edu

## Abstract

Genetic compensation is a common phenomenon in zebrafish in response to genetic alterations. Differences between genetic and morpholino-mediated zebrafish models of human diseases have led to significant difficulties in phenotypic interpretation and translatability. One form of compensation is the maternal deposit of mRNAs and proteins to the oocyte that supports developmental processes before zygotic genome activation. In this study, we generated a zebrafish model of severe congenital muscular dystrophy (CMD) by targeting *protein O-mannose N-Acetylglucosaminyltransferase 2* (*pomgnt2*), a maternally provided gene that maintains cell-extracellular matrix interactions through glycosylation and leads to congenital muscular dystrophy when mutated. Zygotic knockouts (ZKOs) retain protein function in the first week post fertilization and survive to adulthood, only developing muscle disease later in life. In contrast, maternal-zygotic KOs (MZKOs) generated from ZKO females develop early-onset muscle disease, reduced motor function, neuronal axon guidance deficits, and retinal synapse disruptions recapitulating features of the human presentation. While assessing transcriptional changes linked to disease progression, the availability of embryos obtained from different breeding strategies also allowed for a direct comparison of ZKOs and MZKOs to define the impact of having a KO mother. We found that offspring from a ZKO mother, independently of genotype, show distinct expression patterns from animals obtained from heterozygous breedings. Some of these changes reflect changes in metabolic function, possibly stemming from maternal metabolic disruption. These findings will not only be applicable for other CMD models targeting maternally provided genes, but also provide new insight into modeling disease using maternal-zygotic mutants.

which permits unrestricted use, distribution, and reproduction in any medium, provided the original author and source are credited.

**Data availability statement:** All the RNAseq data was deposited on GEO with accession GSE314061. All other relevant data are in the manuscript and its Supporting information files.

**Funding:** This work was funded by the National Institute of Neurological Disorders and Stroke (NINDS, grant R01NS109149), the Robert Wood Johnson Foundation (RWJF, grant #74260), and a CureCMD Pilot Grant all to M.C.M. Salaries for K.F., S.M., N.B., and L.R.C. were partially funded by the NINDS and RWJF grants. S.M. and L.R.C. were also partially funded by the CureCMD grant. URL: NINDS: https://www.ninds.nih.gov/ RWJF: https://www.rwjf.org/ CureCMD: https://www.curecmd.org/ The funders had no role in the study design, data collection and analysis, decision to publish, or preparation of the manuscript.

**Competing interests:** The authors have declared that no competing interests exist.

## Author summary

Generating zebrafish genetic mutants that exhibit similar features to human diseases can be complicated. Unlike humans, zebrafish are highly equipped to compensate for genetic mutations. As a result, mutant zebrafish are often healthy or show signs of disease much later in their lifespan. In this study, we generated a zebrafish strain with mutations in a gene that causes congenital muscular dystrophy, but due to compensation from healthy female parents, the mutant offspring do not show signs of disease until adulthood. Mutant female parents, however, cannot compensate, so their mutant offspring have disease early in life, like humans. Using mutant females to generate mutant offspring helped us study how the disease progresses in the muscles and nervous system. We also showed that this approach could impact the metabolism of these offspring compared to the offspring of healthy females. This finding is highly important, as many other mutant zebrafish designed to model human genetic disorders show the same compensation. Therefore, fully understanding this compensation will also advance our understanding of other zebrafish genetic models of disease.

## Introduction

Genetic compensation in zebrafish is a well-documented phenomenon that has profound implications for the use of zebrafish as a model system in studying developmental processes and human diseases [1]. The basis for genetic compensation arises from many sources, such as genome duplication in a teleost ancestor that led to additional copies of genes that are critical for embryonic development, as well as altered expression of non-duplicated, yet functionally similar genes that can mask the effect of genetic alterations [2–5]. As such, there has been considerable discrepancy between the phenotypes observed in fish following morpholino oligonucleotide (MO)-mediated knockdown strategies (morphants) and stable knock-out (KO) strains, leading to complications in phenotype interpretations [1,6–9].

In addition to classic genetic compensation, another source of phenotypic discrepancy between morphants and KOs stems from mRNAs and proteins deposited from the mother into the yolk of the offspring to direct embryonic development through the maternal to zygotic transition (MZT) [10,11]. Many of these gene products have been identified through maternal effect and gynogenetic screens and were found to be essential for critical processes in embryogenesis such as egg activation, cell polarity, cleavage, and body axis patterning [12–14]. In mutant zebrafish models of human diseases, such as CHARGE syndrome and scoliosis, maternal transcripts have been found to mask the impact of zygotic gene loss requiring the use of KO mothers to deplete maternal mRNAs for full phenotypic presentation [15,16]. Here, we present a zebrafish KO strain for *protein O-linked mannose N-acetylglucosaminyltransferase 2* (*pomgnt2*) as a novel *in vivo* model of congenital neuromuscular disease and leverage this model to define the impact of maternal gene products and declining health in KO mothers.

*POMGNT2* encodes a glycosyltransferase enzyme involved in the glycosylation of α-dystroglycan (α-DG), the extracellular glycoprotein in the dystrophin-glycoprotein complex (DGC) [17,18]. α-DG is bound to its transmembrane subunitβ-DG originating from the same precursor protein and links intracellular dystrophin to ligands in the extracellular matrix (ECM). These extracellular interactions are mediated through an elongated functional glycan, termed matriglycan [19,20]. Matriglycan is initiated via an O-linked mannose (O-Man) on the mucin domain of α-DG by the Protein O-mannosyltransferase 1 and 2 (POMT1, POMT2) complex and then extended by seven glycosyltransferases that add different sugars in a specific order. POMGNT2 catalyzes the addition of an N-acetylglucosamine to the O-Man and is selective for matriglycan over other O-Man initiated chains on α-DG [17,21]. Biallelic variants in any of these glycosyltransferases cause a heterogeneous group of congenital muscular dystrophies (CMDs), termed dystroglycanopathies. Loss of function (LOF) variants in either *POMT1–2* or *POMGNT2* cause Walker Warburg Syndrome (WWS), the most severe dystroglycanopathy that is associated with profound brain and eye malformations including cobblestone lissencephaly, hydrocephalus, cerebellar hypoplasia, and retinal defects [22–24].

Due to a rodent-specific essential role of α-DG in post-implantation, mouse KO models of multiple genes involved in α-DG glycosylation lead to embryonic or perinatal lethality, hindering studies of disease progression beyond birth [25–31]. As such, zebrafish have emerged as a prominent model of dystroglycanopathy. However, zebrafish KOs disrupting α-DG glycosylation present with the same discrepancy between MO and KO models that are prevalent in the field. KOs for *dystroglycan* (*dag1*) itself, also known as *patchytail,* and another glycosyltransferase involved in matriglycan assembly, *fukutin-related protein* (*fkrp*), show muscle disease and lethality within 10 dpf [32,33]. In contrast, KOs for *pomt2* and *protein O-linked mannose N-acetylglucosaminyltransferase 1* (*pomgnt1*) have delayed onset months post fertilization [34,35]. A recent study by our group reconciled these differences in a model for *POMT1* LOF, showing that maternally provided Pomt1 maintained α-DG glycosylation and masked early developmental phenotypes in zebrafish [36]. We found the same to be true in developing a model of *POMGNT2* LOF.

While generating maternal-zygotic mutants through zygotic mutant females has been widely undertaken to study the effect of maternally provided gene loss, there has not been as much investigation into the effects on offspring outcomes that may arise due to the zygotic mutations in female parents. However, when performing transcriptomic analyses to investigate disease progression, we also defined the differences between zygotic *pomgnt2* KO (ZKO) embryos that initially retain α-DG glycosylation and maternal-zygotic KOs (MZKOs) that do not. We revealed distinct correlations in gene expression patterns reflecting changes in metabolic function in MZKOs as well as their heterozygous siblings. These findings demonstrate how muscle wasting and declining health in zygotic mutant females may lead to physiological changes in their progeny and will have immediate applicability for the generation of novel zebrafish models of CMD caused by mutations in genes that are maternally provided to the embryo.

## Results

### Loss of zygotic *pomgnt2* leads to adult-onset muscle phenotypes

To generate a *pomgnt2* KO strain*,* we used CRISPR-Cas9-induced nonhomologous end joining (NHEJ) with guide RNAs (gRNAs) targeting each of the two coding exons. We identified multiple frameshifts that disrupted *pomgnt2* in F1 founders. Deletions near the protospacer adjacent motif (PAM) sequence in exon 1 led to the usage of an alternative start codon, but we identified a 13 bp insertion (NM_001012384: c.17insTAAAATAAGGCTA, p.Cys6fs) with an additional 4 bp deletion (NM_001012384: c.32_35del, p.Pro11fs) that also disrupted the alternative start site. In addition, we found a 7 bp deletion in exon 2 in the glycosyltransferase domain (NM_001012384: c.713_719del, p.Ser238fs), and founders carrying both variants on the same allele (Fig 1A). There are no suitable antibodies to test for Pomgnt2 protein expression in zebrafish. We tested KO embryos from heterozygous crosses (HetxHet) for nonsense mediated mRNA decay via qPCR and found that the mutated mRNA was present (S1A Fig). However, we confirmed Pomgnt2 LOF through western blotting using the α-DG glyco-specific antibody clone IIH6C4 on glycoprotein-enriched lysates. We found that α-DG glycosylation was absent in

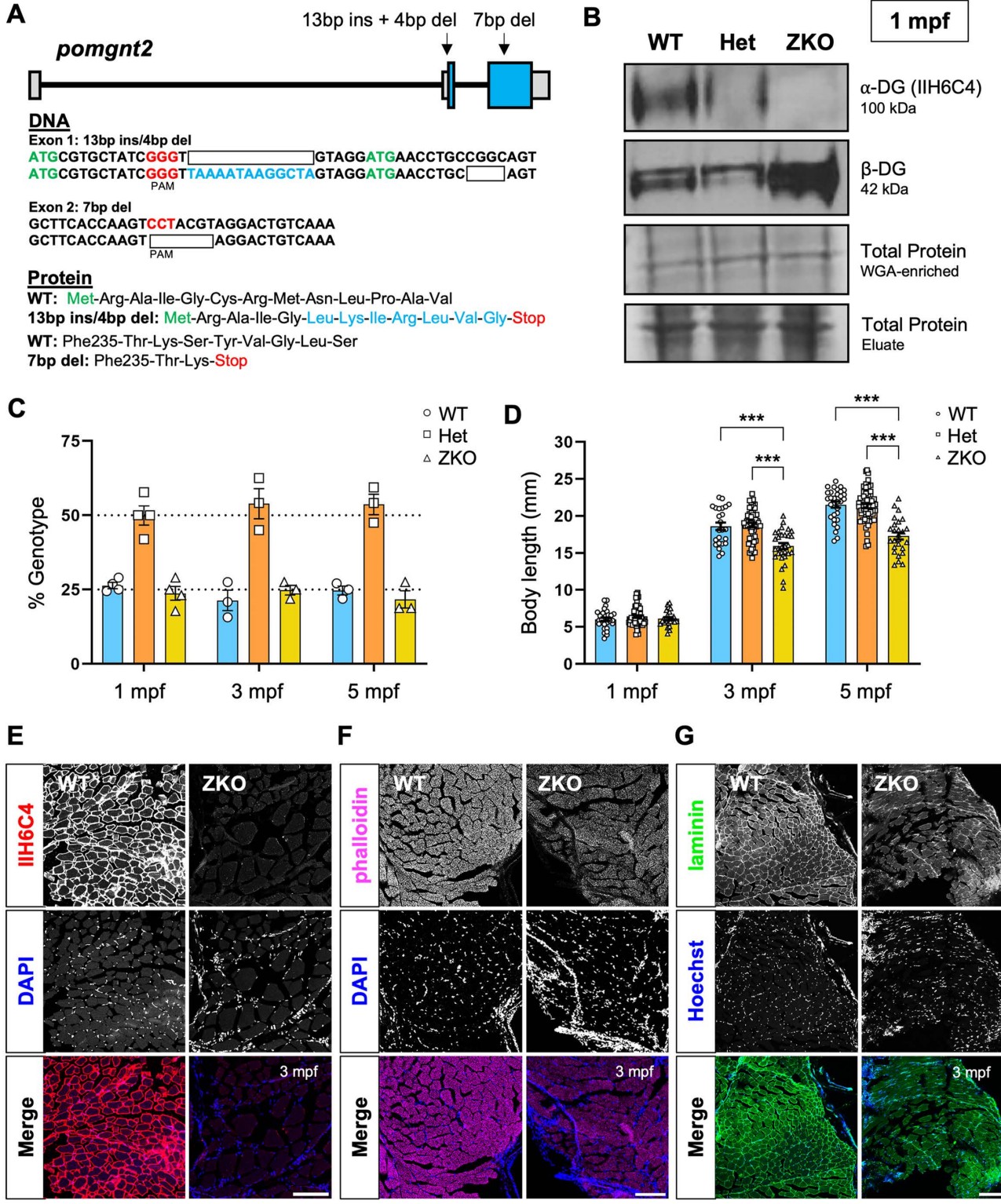

**Fig 1. Generation of the *pomgnt2* line and adult-onset phenotypes in ZKOs. A**: Mutation schematic showing indels in exons 1 and 2 induced through CRISPR-Cas9 nonhomologous end joining. **B**: Western blot showing complete loss of α-DG glycosylation labeled via the glyco-specific IIH6C4 antibody in ZKOs at 1 mpf, even when glycoprotein-enriched lysate is in excess as shown through β-DG protein levels. **C**: Survival analysis showing

ZKOs surviving in Mendelian ratios through 5 mpf. Expected survival rates of 25% for WTs and ZKOs and of 50% for Hets is indicated by dotted lines. **D**: Body length measurements showing comparable body length between ZKOs and their siblings at 1 mpf, but a significant reduction in ZKOs at 3 and 5 mpf (***p < 0.001). **E–G**: Images of transverse muscle sections stained for α-DG glycosylation (IIH6C4) **(E)**, F-actin filaments **(F)**, and laminin **(G)** showing fiber separation, disorganization, and possible fibrosis in ZKOs (20X magnification, scale bars: 100 μm).

all three KO lines at 1 month post-fertilization (mpf), while β-DG, which is produced by the same transcript and indicated dystroglycan enrichment, was present in excess (Figs 1B and S1B, S1C). We used the line disrupting both exon 1 and exon 2 for further study.

To determine how loss of *pomgnt2* impacts the overall health of the zebrafish, we examined survival, gross morphology, and muscle structure. We found that zygotic KOs (ZKOs) from heterozygous crosses (Het X Het) survive in Mendelian genotypic ratios into early adulthood (1 mpf: WT 26.35 ± 1.04%; Het 49.93 ± 3.23%; ZKO 23.72 ± 2.34%; N = 4 cohorts, n = 134; 3 mpf: WT 21.32 ± 3.44%; Het 53.89 ± 5.05%; ZKO 24.79 ± 1.63%; N = 3 cohorts, n = 120; 5 mpf: WT 24.65 ± 1.51%; Het 53.68 ± 3.44%; ZKO 21.67 ± 2.92%; N = 3 cohorts, n = 120) (Fig 1C). In addition, no differences in body length were observed at 1 mpf (Fig 1D), but ZKOs were significantly smaller than their WT and Het siblings at 3 and 5 mpf (1 mpf: WT: 6.04 ± 0.23 mm, n = 30; Het: 6.41 mm ± 0.18, n = 53; ZKO: 6.11 mm ± 0.20, n = 26; N = 3 cohorts. p > 0.9999; 3 mpf: WT: 18.61 mm ± 0.51, n = 24; Het: 18.76 mm ± 0.29, n = 57; ZKO: 15.93 mm ± 0.41, n = 30; ***p < 0.001. 5 mpf: WT: 21.53 mm ± 0.40, n = 30; Het: 21.33 mm ± 0.28, n = 62; ZKO: 17.29 mm ± 0.47, n = 26; N = 3 cohorts. ***p < 0.001) (Fig 1D).

One of the main features of dystroglycanopathy is loss of muscle integrity which mirrored the delayed growth in ZKOs. Muscle fibers stained with fluorescently conjugated phalloidin were organized and uniform in ZKOs at 1 mpf, comparable to WT (S1D Fig). However, general signs of muscle disease were present at 3 mpf and visualized through fluorescent staining for α-DG glycosylation, F-actin filaments, and laminin, showing muscle fiber separation, size variability, disorganization, and nuclear staining patterns suggestive of fibrosis (Fig 1E–1G). While survival was only formally assessed through 5 mpf, ZKOs could be maintained up to 1 year of age at lower stocking densities to limit food competition. At this time, muscle disease in ZKOs was significantly advanced with evidence of severe fibrosis (S2A Fig) and disrupted locomotor function (S2B–S2D Fig). Taken together, these findings indicate that ZKOs exhibit prolonged survival but have progressive muscle disease in adulthood.

## Loss of maternal and zygotic pomgnt2 unmasks early developmental phenotypes

While loss of α-DG itself in *dag1* ZKOs leads to severe phenotypes during the first two weeks post fertilization [32], the mild phenotypic presentation in *pomgnt2* ZKOs mirrored disease progression in *pomt1* ZKOs where maternally provided *pomt1* masked developmental phenotypes [36]. *pomgnt2* mRNA has been detected in the embryo at developmental stages before the MZT in RNA sequencing studies [37]. We performed independent qPCR validation between 0 and 96 hours post fertilization (hpf) showing that it is maternally provided (0 hpf: 1.05 ± 0.23; 24 hpf: 0.04 ± 0.004; 48 hpf: 0.13 ± 0.01; 72 hpf: 0.16 ± 0.03; 96 hpf: 0.37 ± 0.05) (Fig 2A). We confirmed that *pomgnt2* ZKOs retain α-DG glycosylation in embryos and larvae by examining IIH6C4 immunostaining at 7 dpf (Fig 2B). We then bred ZKO females with Het males resulting in progeny with an expected ratio of 50% maternal *pomgnt2* Hets (MHets) and 50% maternal zygotic KOs (MZKOs). Here, we found that α-DG glycosylation was eliminated in MZKOs at 7 dpf (Fig 2B), supporting our hypothesis that maternally provided *pomgnt2* was the most likely source of residual glycosylation in ZKOs.

MZKO survival progressively declined between 10 and 14 dpf and the majority of MZKOs died by 28 dpf (7 dpf: MHet 49.57 ± 0.43%; MZKO 50.40 ± 0.40%; N = 3 cohorts. 10 dpf: MHet 54.93 ± 4.75%; MZKO 45.07 ± 4.75%; N = 3 clutches. 14 dpf: MHet 62.18 ± 5.13%, n = 42; MZKO 37.83 ± 5.13%; N = 4 clutches. 28 dpf: MHet = 87.98 ± 4.70%,

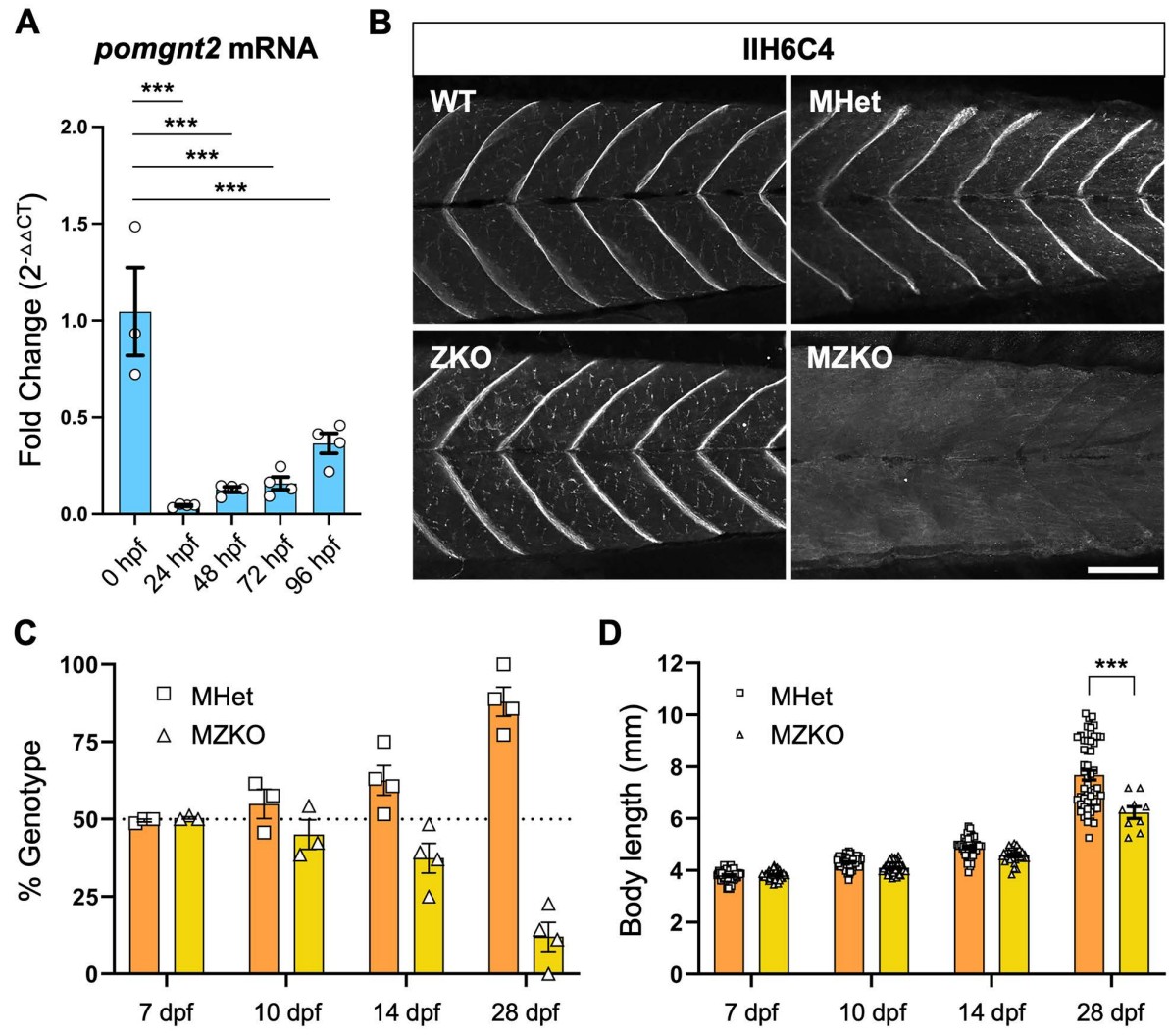

**Fig 2. Developmental phenotypes caused by elimination of maternal *pomgnt2*. A**: qPCR analysis showing high *pomgnt2* mRNA expression in oocytes and increasing zygotic expression between 24 and 96 hpf. (***p < 0.001) **B**: Immunofluorescent staining of muscle showing that ZKOs have residual α-DG glycosylation due to maternally provided *pomgnt2*, but this is depleted in MZKOs without maternal *pomgnt2* (10X magnification, scale bar: 100 μm). **C**: Survival analysis showing that MZKOs begin to deviate from the Mendelian 50/50 survival ratio between 10-14 dpf, with most MZKOs dead by 28 dpf. **D**: Body length measurements showing a trend toward reduction in MZKOs at 14 dpf and a significant reduction at 28 dpf (***p < 0.001).

MZKO = 12.03 ± 4.70%; n = 61, N = 4 clutches) (Fig 2C). Data from each timepoint were examined collectively by Chi-square analysis, indicating a statistically significant deviation in survival beginning at 14 dpf that increases drastically by 28 dpf (14 dpf: MHet n = 59, MZKO n = 39, $\chi^2$ = 4.08, df = 1, *p = 0.04. 28 dpf: MHet n = 52, MZKO n = 9, $\chi^2$ = 29.4, df = 1, ***p < 0.001) (S1 Table). Body length did not differ at 10 and 14 dpf, but the few MZKOs that survived to 28 dpf were significantly smaller than their MHet siblings (Fig 2D) (7 dpf: MHet 3.82 ± 0.03 mm, n = 67; MZKO 3.83 ± 0.03 mm, n = 42; p > 0.9999. 10 dpf: MHet 4.32 mm ± 0.03, n = 66; MZKO 4.11 ± 0.03 mm, n = 55; N = 3 clutches; p = 0.1611. 14 dpf: MHet 4.90 ± 0.06 mm, n = 42, MZKO 4.60 ± 0.06 mm, n = 30; N = 3 clutches; p = 0.0857; 28 dpf: MHet 7.68 ± 0.19 mm, n = 52; MZKO 6.24 ± 0.23 mm, n = 9; N = 4 clutches; ***p < 0.001). Collectively, these assessments show that removing maternal *pomgnt2* mRNA unmasks developmental phenotypes in MZKO larvae.

## MZKOs show dystroglycanopathy phenotypes within the first 2 weeks post-fertilization

Severe dystroglycanopathy presents as loss of mobility and muscle integrity, retinal abnormalities, and neuronal axon guidance deficits in fish models [32–36]. To determine the onset of dystroglycanopathy phenotypes in MZKOs, we performed multiple muscle integrity and function analyses. Locomotor activity analysis using automated tracking in open swimming trials showed that at 5 dpf, prior to the onset of differences in body size, MZKOs already exhibited profound deficits in swimming behaviors. Significant differences were found in total swimming distance (Fig 3A) (MHet: 375.2±12.13 cm, n=79; *MZKO*: 226.6±5.96 cm, n=90; ***p<0.001), average velocity (Fig 3B) (MHet: 0.21±0.007 cm/s, n=79; MZKO: 0.13±0.003 cm/s, n=90; ***p<0.001), maximum velocity (S3A Fig) (MHet: 7.42±0.24 cm/s, n=79; MZKO: 6.16±0.23 cm/s, n=90; ***p<0.001), and maximum acceleration (S3B Fig) (MHet: 191.4±7.07 cm/s$^2$, n=79; MZKO: 155.6±6.83 cm/s$^2$, n=90; ***p<0.001).

We further leveraged locomotor behavior as a readout of muscle function at 7, 10, and 14 dpf with a larger swimming arena to help define the course of disease progression. A significant reduction in distance (7 dpf: MHet 297.4±15.95 cm, n=59; MZKO 200.0±18.19 cm, n=36; 10 dpf: MHet 291.5±8.56 cm, n=66; MZKO 177.1±9.86 cm, n=54; 14 dpf: MHet 388.3±12.36 cm, n=68; MZKO 257.7±14.94 cm, n=35) and average velocity (7 dpf: MHet 0.30±0.013 cm/s, n=59; MZKO 0.23±0.015 cm/s, n=36; 10 dpf: MHet 0.26±0.007 cm/s, n=66; MZKO 0.18±0.007 cm/s, n=54; 14 dpf: MHet 0.33±0.011 cm/s n=68; MZKO 0.23±0.012 cm/s, n=35) was noted at all three time points (S3C, S3D Fig), but there was an increasing difference in maximum velocity and maximum acceleration between MHets and MZKOs between 7 and 14 dpf suggesting a decline in muscle strength after 7 dpf (Fig 3C, 3D) (Maximum Velocity: 7 dpf MHet 10.36±0.40 cm/s, MZKO 8.60±0.68 cm/s, **p=0.0022; 10 dpf MHet 10.47±0.31 cm/s, MZKO 6.84±0.33 cm/s, ***p<0.001; 14 dpf MHet 11.99±0.49 cm/s, MZKO 8.33±0.52 cm/s, ***p<0.001; Maximum Acceleration: 7 dpf MHet 266.43±12.32 cm/s$^2$, MZKO 201.30±16.51 cm/s$^2$, *p=0.0321; 10 dpf MHet 268.12±9.52 cm/s$^2$, MZKO 162.04±9.44 cm/s$^2$, ***p<0.001; 14 dpf MHet 305.02±13.15 cm/s$^2$, MZKO 195.88±14.36 cm/s$^2$, ***p<0.001).

Despite the presence of motor deficits at 7 dpf, integrity of actin filaments in muscle fibers labeled with fluorescently conjugated phalloidin (Fig 3E) and of the laminin basement membrane at the myotendinous junctions (MTJs) (Fig 3F) was generally unaffected in MZKOs, with the exception a few occasional detached myofibers (Fig 3E). Interestingly, immunostaining for slow-twitch muscle fibers using myosin heavy chain 1A (Myh1a – F59 clone) revealed sparse detachment at the MTJ in some myotomes, similar to the early fiber detachment found in Duchenne Muscular Dystrophy (DMD) and LAMA2-related dystrophy models (Fig 3E) [38,39]. By 10 dpf, muscle disease in MZKOs had progressed rapidly, with several myotomes displaying detached, atrophied, and disorganized muscle fibers, disrupted and split MTJs, and complete deterioration of Myh1a-positive fibers (Fig 3F).

We next examined the structure of the neuromuscular junctions (NMJs), as Agrin, a major contributor to acetylcholine receptor (AChR) clustering, is a well-characterized α-DG binding partner [40–42]. We visualized AChRs using fluorescently labeled α-bungarotoxin (α-BTX) and motor neuron terminals via immunostaining for Synaptic vesicle protein 2 (SV2) at 10 dpf (Fig 4A). No significant differences were noted between MHets and MZKOs in the myotome in α-BTX intensity or density and in SV2 puncta density (Fig 4B–4D) (α-BTX intensity: MHet: 0.1254±0.006, n=12; MZKO: 0.1122±0.007, n=8; p=0.1813. α-BTX puncta density: MHet: 1.39±0.11 puncta/100 μm$^2$, n=12; MZKO: 1.58±0.21 puncta/100 μm$^2$, n=8; p=0.3788. SV2 puncta density: MHet: 2.05±0.10 puncta/100 μm$^2$, n=12; MZKO: 1.76±0.11 puncta/100 μm$^2$, n=8; p=0.0819). However, there was a significant reduction in α-BTX/SV2 colocalization, indicating a disruption in NMJ synapse integrity that could impact muscle innervation (MHet: 0.7562±0.013, n=12; MZKO: 0.6987±0.007, n=8; **p=0.003) (Fig 4E). AchR cluster fragmentation was more evident at the MTJ, where we observed a significant reduction in α-BTX intensity paired with an increase in α-BTX puncta density (Fig 4B', 4C') (α-BTX intensity: MHet: 0.1548±0.006, n=12; MZKO: 0.1110±0.005, n=8; ***p<0.001. α-BTX puncta density: MHet: 4.78±0.08 puncta/100 μm$^2$, n=12; MZKO: 5.86±0.10 puncta/100 μm$^2$, n=8; ***p<0.001). SV2 puncta density was not significantly different (Fig 4D') (MHet: 3.77±0.24 puncta/100 μm$^2$, n=12; MZKO: 3.34±0.17 puncta/100 μm$^2$, n=8; p=0.2045). Lastly, α-BTX/SV2 colocalization

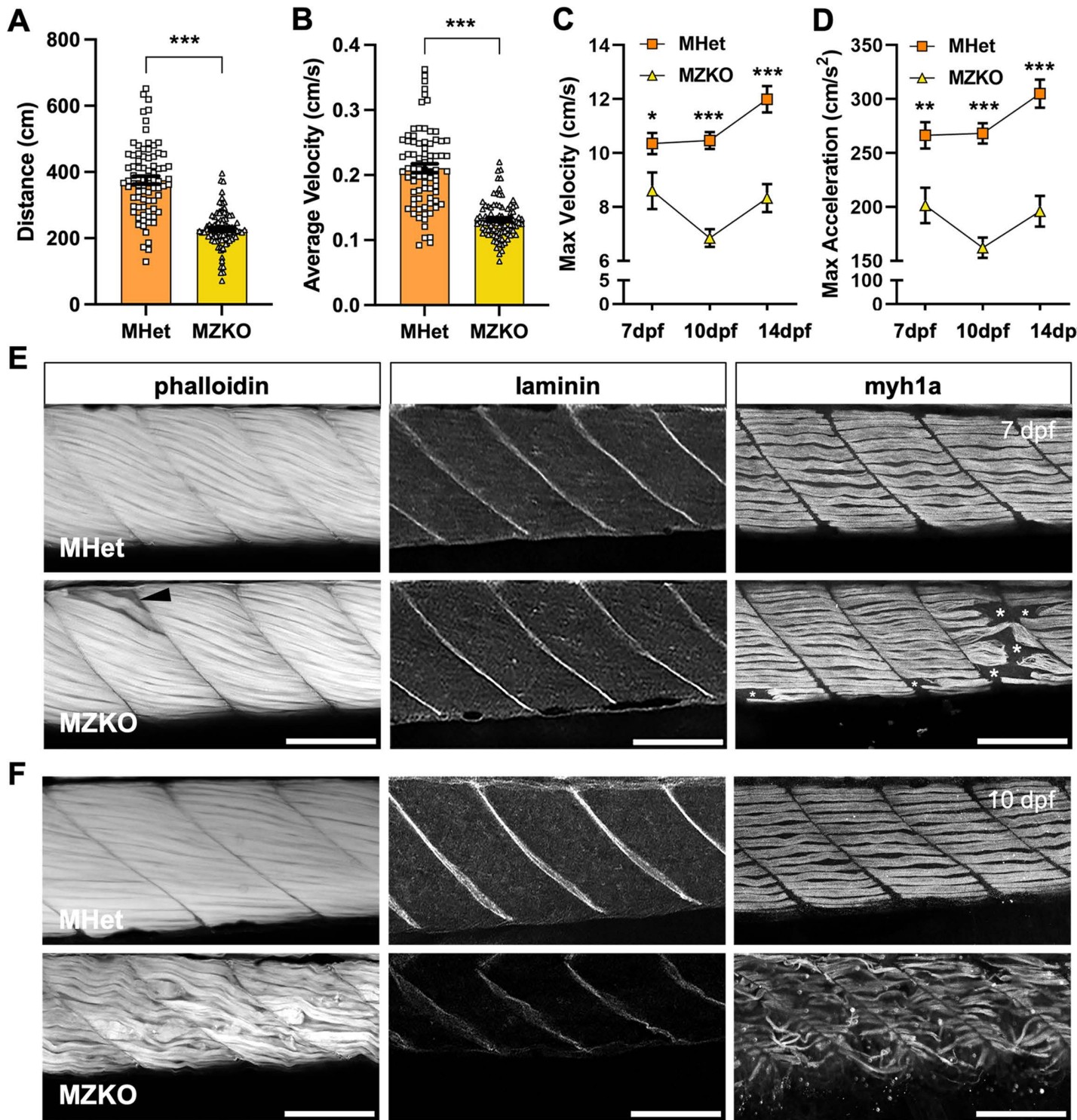

**Fig 3. Early onset muscle and motor phenotypes in MZKOs. A, B**: Analysis of locomotor function at 5 dpf showing significant reductions in distance (**A**) and average velocity (**B**) in MZKOs (***$p < 0.001$). **C, D**: Analysis of locomotor function between 7-14 dpf showing that MZKOs have significant reductions in maximum velocity (**C**) and acceleration (**D**) at all three time points compared to their MHet siblings but show an overall decline between 7-14 dpf (*$p < 0.05$, **$p < 0.01$, ***$p < 0.001$). **E**: Fluorescent staining of muscle at 7 dpf showing overall uniform muscle organization with a few detached fibers (**left,** arrowhead), normal laminin signal at the myotendinous junctions (**middle**), and detachments within the slow twitch fiber layer (**right,** asterisks)

(20X magnification, scale bars: 100 µm). **F**: Fluorescent staining of muscle at 10 dpf showing overall deterioration of muscle fibers (**left**), diffuse and disrupted laminin signal at the myotendinous junctions (**middle**), and uniform degeneration of the slow twitch fiber layer (**right**) (20X magnification, scale bars: 100 µm).

showed a larger reduction at the MTJ than in the myofibers (Fig 4E') (MHet: 0.7900±0.01, n=12; MZKO: 0.6446±0.01, n=8; ***p<0.001). Taken together, these data indicate that loss of *pomgnt2* affects the NMJs, with AchR fragmentation at the MTJ in end plates primarily innervated by secondary motor neurons.

Retinal photoreceptor synapse loss is another hallmark feature of dystroglycanopathies recapitulated in zebrafish models [34,35]. We examined how the 10 dpf retina is impacted by immunostaining with an anti-Arrestin-3a (Arr3a) antibody (zpr1 clone) to outline both the outer segment and pedicles of cone photoreceptors, and with an anti-Synaptophysin (Syp) antibody to identify presynaptic vesicles at ribbon synapses in the photoreceptor pedicles. Syp staining showed occasional discontinuities and reduced intensity in the outer plexiform layer (OPL) (MHet: 0.1707±0.018, n=8; MZKO: 0.0778±0.023, n=6; **p=0.0069) (Fig 5A, 5B). In addition, photoreceptor cell bodies in the outer nuclear layer (ONL) were often less organized and occasionally protruded into the OPL, though the outer segment of the cones showed comparable organization between MHets and MZKOs indicating that these disruptions to the inner retinal layers have not yet led to photoreceptor death. Horizontal cell disruptions had been noted in *pomt1* MZKOs [36], but organization of nuclei visualized with DAPI lining the upper border of the inner nuclear layer (INL) and processes immunostained by calbindin were normal in *pomgnt2* MZKOs (S4 Fig).

Finally, we examined the structure of the optic chiasm, as α-DG plays a role in axon guidance through interactions with the Slit family of axon guidance cues that facilitate midline crossing [43,44]. Using the zn-8 antibody to label retinal ganglion cell axons, we found evidence of defasciculation that was most prominent following decussation of the optic nerves, as was observed in *pomt1* and *slit2* mutants (Fig 5C) [36,45]. This suggests that loss of *pomgnt2* impacts the retina and axon guidance in a manner similar to other genes involved in α-DG glycosylation. Overall, MZKOs recapitulate features of dystroglycanopathy and closely resemble other severe CMD models.

## Loss of maternal *pomgnt2* leads to widespread differences in gene expression

To investigate both the molecular changes involved in disease progression in MZKOs and potential sources of compensation in ZKOs, we performed transcriptomic analyses at different ages. To define changes associated with the severe disease state when MZKOs begin to die, we conducted bulk RNA sequencing (RNA-seq) analyses on samples obtained from whole 10 dpf larvae, comparing them with MHet siblings (S1 File). We identified 959 differentially expressed genes (DEGs), with 810 significantly downregulated and 149 significantly upregulated DEGs (padj<0.01; FC<0.7) (Fig 6A). Gene Ontology (GO) analysis showed a large number of enriched biological processes (BPs), many of which were redundant and required curation through analysis of overlapping genes. The most highly enriched BPs, molecular functions (MFs) and cellular components (CCs) were related to muscle function, metabolism, and regulation of proteolytic activity among the downregulated DEGs (Fig 6B). In contrast, increased expression of *jacalin* family genes led to GO enrichment in mannose binding among the upregulated DEGs. This analysis was further refined using local networks defined by protein-protein interactions in STRING, which revealed several additional downregulated networks. These included larger interconnected networks including muscle contraction and actin filament capping, in addition to complement activation, coagulation, fatty acid binding, phospholipid metabolism, and LDL remodeling, the latter of which was driven by reduced expression of many apolipoprotein transcripts (*apoa1a, apoa1b, apoea, apoa4a, apoa4b.1, apoa4b.2, apoc2, apom*) (Fig 6C). Additional distinct networks of downregulated genes included those involved in glycolysis and gluconeogenesis, glycerophospholipid metabolism, sterol/cholesterol biosynthesis, and amino acid (glycine, serine, threonine) metabolism (Fig 6D). These findings suggest that MZKOs are in a severe state of muscle disease with global disruptions in metabolic processes.

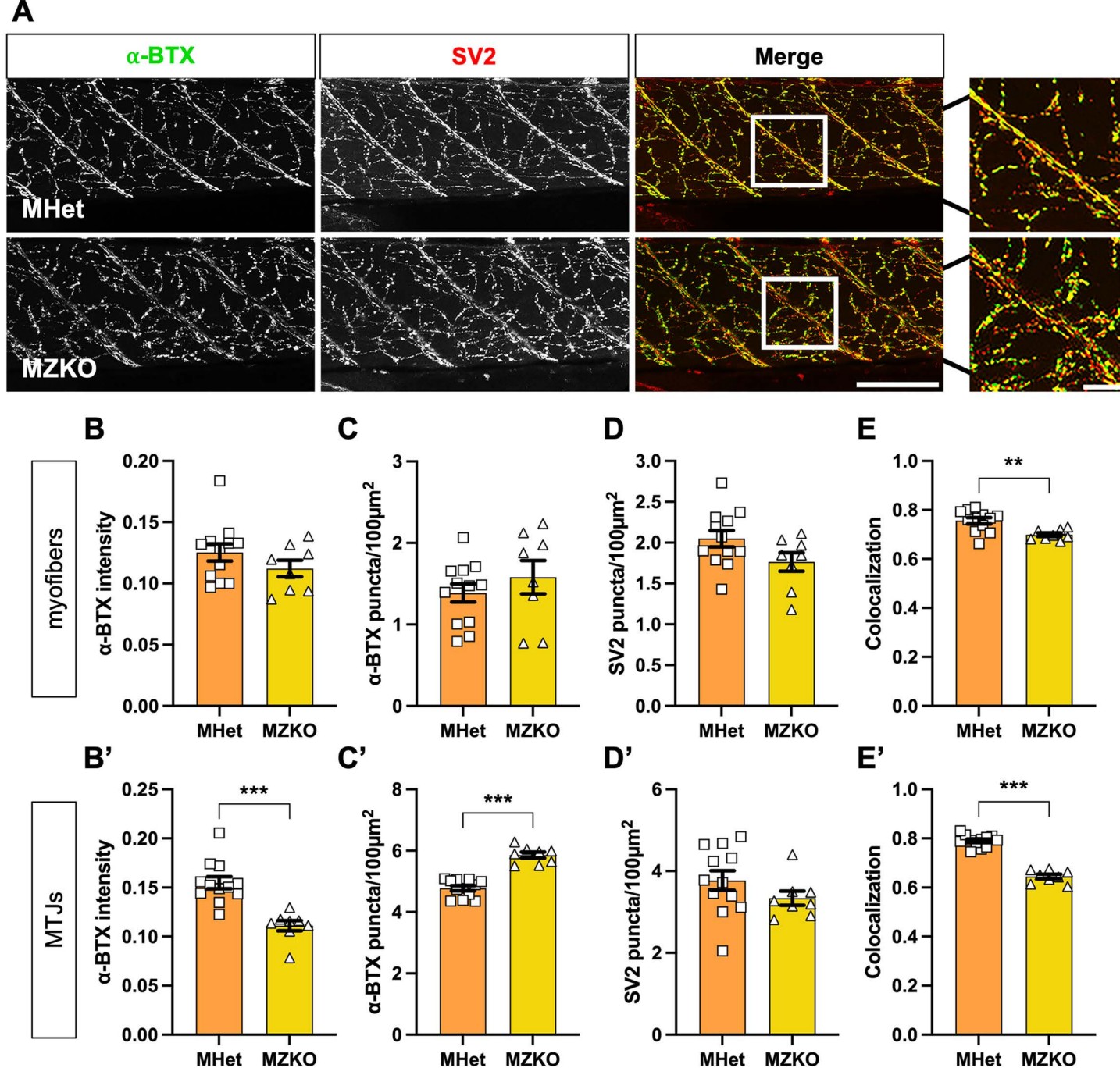

**Fig 4. Evaluation of neuromuscular junction integrity in MZKOs. A:** Maximum intensity projections of muscle stained with α-bungarotoxin (BTX) to label Acetylcholine receptors on the sarcolemma (**left**), anti-SV2 antibody to label motor neuron terminals (**middle**), and merged images to evaluate colocalization (**right**) (20X magnification, scale bars: 100 μm for main panel, 20 μm for insets). **B–D**: Quantifications of NMJs within the myotomes showing no significant differences in α-BTX intensity (**B**), α-BTX puncta density (**C**), or SV2 puncta density (**D**), but a significant reduction in colocalization (**E**) in MZKOs (**p = 0.004). **B'–D'**: Quantifications of neuromuscular junctions at the myotendinous junctions showing significantly reduced α-BTX intensity (**B'**), significantly increased α-BTX puncta density (**C'**), no change in SV2 puncta density (**D'**), and a significant reduction in colocalization (**E'**) (***p < 0.001).

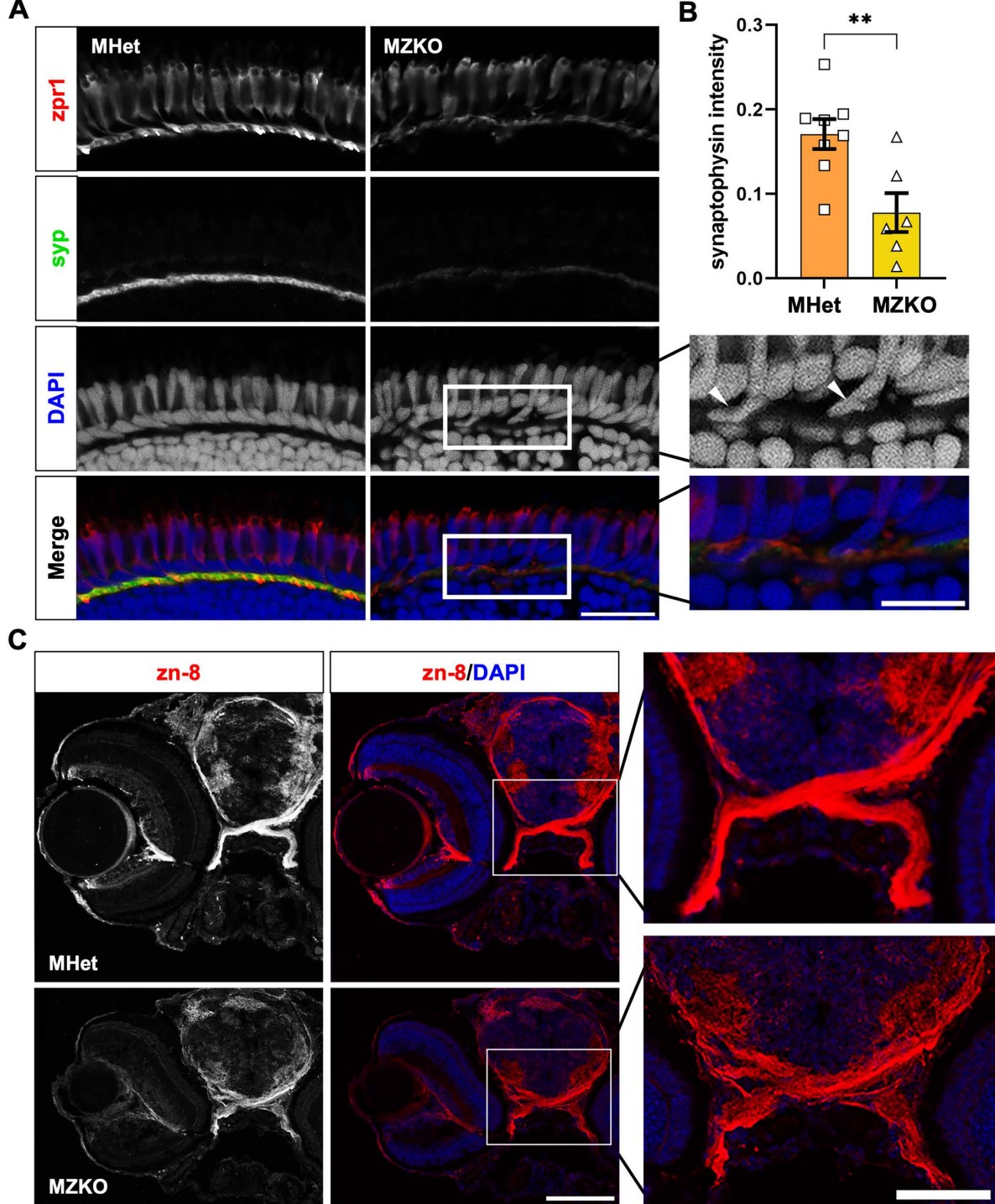

**Fig 5. Retinal synapse formation and axon guidance deficits in MZKOs. A**: Fluorescent staining of the outer layers of the retina showing comparable zpr1 (Arr3a) staining in the photoreceptor outer segments and pedicles in the outer plexiform layer, but reduced and disrupted synaptophysin (syp) staining, protrusion of photoreceptor cell bodies in the outer plexiform layer (arrowheads), and disorganization of horizontal cells lining the top of the outer nuclear layer (asterisks) suggesting defects to the ribbon synapses (40X magnification, scale bars: 20 µm for main panels, 10 µm insets). **B**: Quantification of synaptophysin intensity in the outer plexiform layer showing a significant reduction in MZKOs (**p = 0.0069). **C**: Fluorescent staining of transverse cryosections with zn-8 and DAPI showing an overall disruption in morphology of the retina with evidence of retinal ganglion cell axon defasciculation in the optic nerves at the chiasm (10X magnification, scale bars: 100 µm for main panels, 20 µm for chiasm insets).

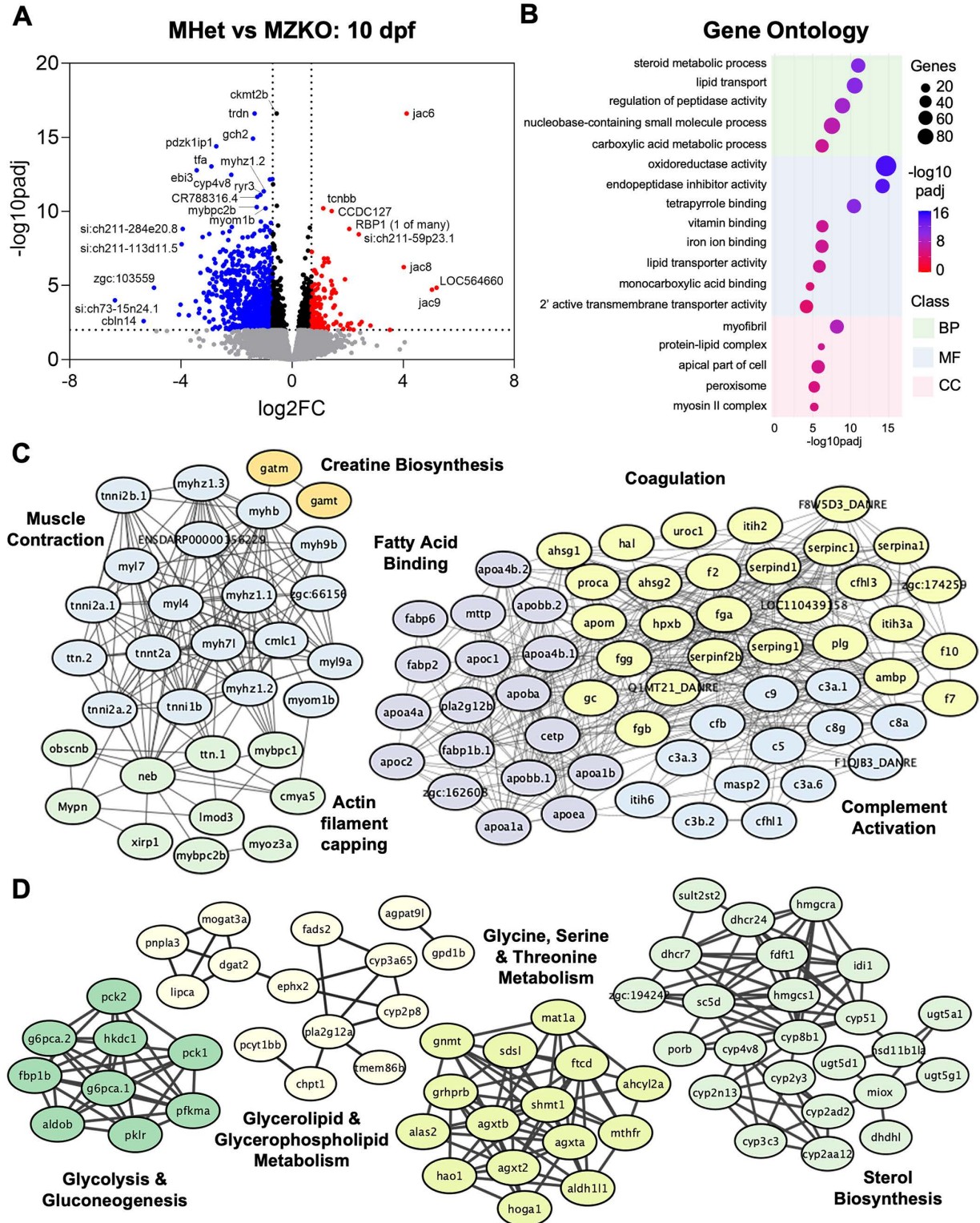

**Fig 6. Differential gene expression analysis in MHets and MZKOs at 10 dpf. A**: Volcano plot showing 810 significantly downregulated DEGs and 149 significantly upregulated DEGs. **B**: Gene Ontology analysis showing various enriched Biological Processes, Molecular Functions, and Cellular Components among the downregulated DEGs pertaining to lipid and cholesterol metabolism and muscle differentiation and function. **D, E**: STRING local networks analysis of interactions among the protein products of the downregulated DEGs showing a large, interconnected network of proteins involved

in muscle contraction and actin filament capping, as well as fatty acid binding, coagulation, and complement activation **(D)**, and smaller networks of proteins involved in glycolysis and gluconeogenesis, glycerophospholipid metabolism, amino acid metabolism (glycine, serine, threonine), and sterol biosynthesis **(E)**.

MZKOs were also tested at an earlier timepoint at 5 dpf when mobility is reduced but muscle integrity is preserved and further compared with the progeny of Het x Het crosses. ZKOs still show WT levels of α-DG glycosylation due to maternal compensation (Fig 2A) and we wanted to test whether additional compensatory changes were present. Transcriptional dysregulation was more modest in ZKOs, than MZKOs (Fig 7A and 7C). However, ZKOs still showed 17 upregulated and 203 downregulated DEGs when compared to WTs (Fig 7A). Among the upregulated genes were the transcription factors *junba, junbb, fosab,* and *egr2a*, which are involved in signaling cascades that promote cell proliferation and differentiation. Notably, genes that act on functional glycan assembly, including *pomt1, pomt2, pomk,* and *b3galnt2* were not differentially expressed, nor was *pomgnt1*, confirming maternally provided *pomgnt2* as the most likely source of residual α-DG glycosylation in ZKOs (S5A Fig). However, the glucuronosyltransferase *b3gat1b* which is involved in the synthesis of a unique sulfated trisaccharide, the HNK-1 epitope, was also upregulated possibly indicating a compensatory response to increase cell-cell and cell-ECM interactions independent of α-DG [46]. Downregulated DEGs belonged to networks involved in cation homeostasis, complement activation, and metabolism of carbon, amino acids, and phospholipids indicating some metabolic disruptions (Fig 7B). A small network of genes involved in the formation of the myosin II complex involved in were also noted among the downregulated DEGs (Fig 7B).

5 dpf MZKOs showed a distinct transcriptional disruption with 159 upregulated and 84 downregulated DEGs compared to MHets (Fig 7C). While there was no significant enrichment for GO BPs, COMPARTMENTS analysis for subcellular localization along with several enriched GO CCs indicated that many upregulated DEGs are present on membranes. Plasma membrane proteins included *sarcoglycan* δ (*sgcd*) and *syntrophin β2* (*sntb2*), which promote membrane stability within the DGC, as well as several *integrins* (*itgb3a, itgb3b*) (Fig 7D), suggesting a compensatory response to stabilize cell-ECM interactions. Further supporting this finding, when we individually examined expression of transcripts encoding dystroglycan and the functional glycan glycosyltransferases as *dag1* was upregulated (p < 0.001***, padj = 0.001**) despite not meeting our strict fold change criteria (S5B Fig). This also paralleled our previous western blots indicating an increase in *β-DG* in glycoprotein enriched ZKO samples where α-DG glycosylation was lost (Figs 1B and S1B, S1C). In parallel, negative regulators of muscle development such as *eif4ebp3l* and *ssh2a* were downregulated while *fbxo32*, which is linked to muscle wasting, was upregulated.

Importantly, despite the frameshift mutations rendering the protein nonfunctional (Fig 1B), *pomgnt2* mRNA did undergo nonsense mediated decay and was not differentially expressed in KO larvae regardless of female parentage (S5A, S5B Fig). This was consistent with our qPCR analyses in ZKOs (S1A Fig). Regardless, the transcriptomic analyses support that MZKOs are in a severe state of muscle and metabolic disease at 10 dpf, and some disease-relevant pathways become dysregulated in both ZKOs and MZKOs at 5 dpf before the full onset of disease phenotypes despite different patterns of transcriptional dysregulation that are consistent with preserved α-DG glycosylation in ZKOs.

## Metabolic disruption correlating with maternal genotype

To further probe whether the DEG expression patterns in the different genotypes at 5 dpf could reveal additional biological changes, we used weighted gene co-expression network analysis (WGCNA) comparing the three genotypes obtained from HetxHet crosses, WT, Het and ZKO, and from KOxHet crosses, MHet and MZKO (S2 File). Following dataset reduction and soft thresholding (S6A, B Fig), this analysis identified seven modules comprised of 1818 genes total whose expression was strongly correlated (S6C, S6D Fig). Four of the seven gene modules were strongly correlated with specific genotypes and crosses (Fig 8A), with a positive correlation indicating increased expression and a negative correlation

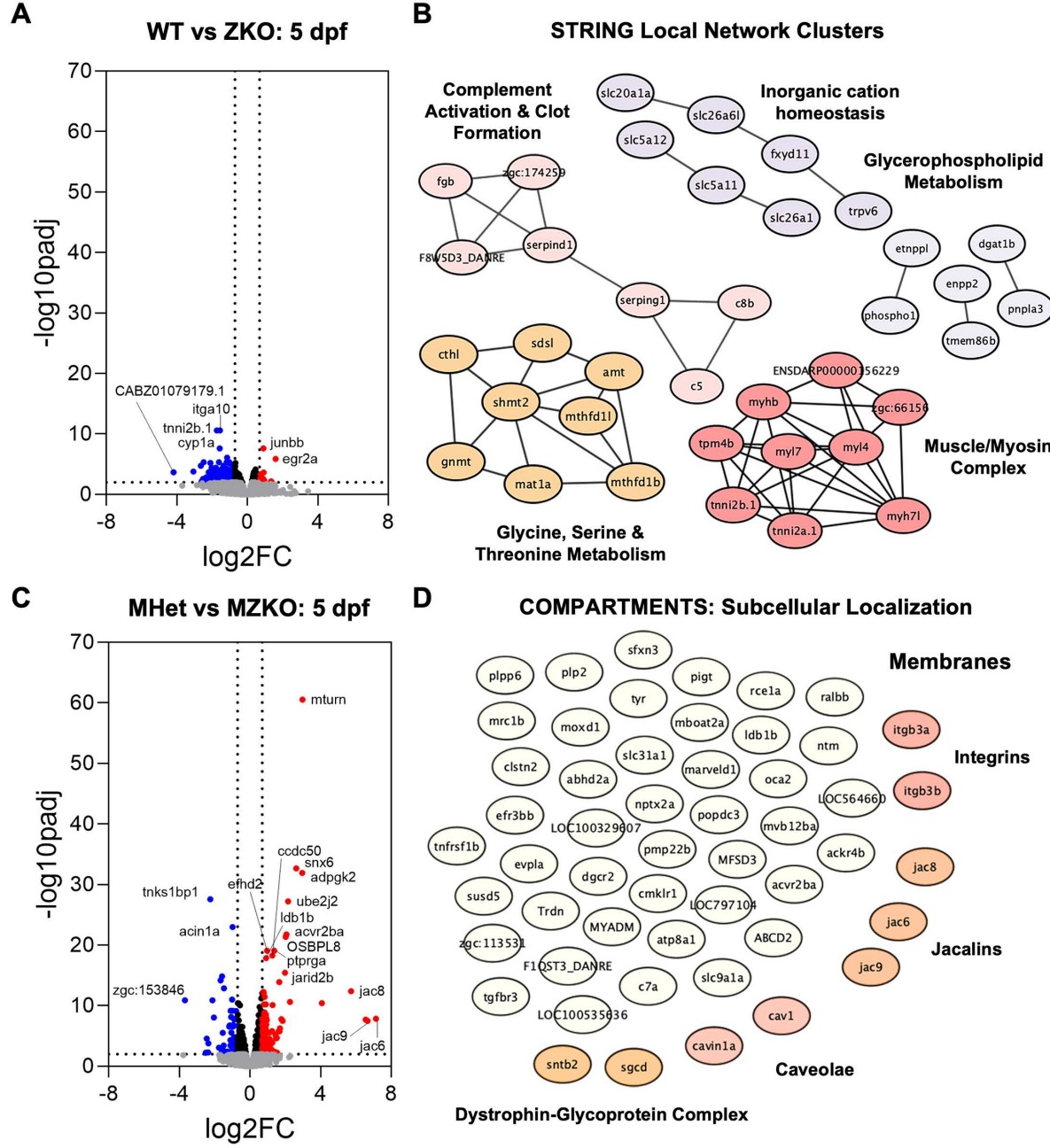

**Fig 7. Differential gene expression analysis at 5 dpf. A**: Volcano plot showing 203 downregulated DEGs and 17 upregulated DEGs in ZKOs compared to their WT siblings. **B**: STRING local networks analysis of downregulated DEGs showing networks of interacting proteins involved in complement activation and clot formation, inorganic cation homeostasis, glycerophospholipid metabolism, amino acid metabolism (glycine, serine, threonine), and muscle function within the myosin II complex. **C**: Volcano plot showing 84 downregulated DEGs and 159 upregulated DEGs in MZKOs compared to their MHet siblings. **D**: Subcellular localization analysis of the downregulated DEGs in **C** through COMPARTMENTS showing an enrichment in genes that encode proteins that are present in the membrane.

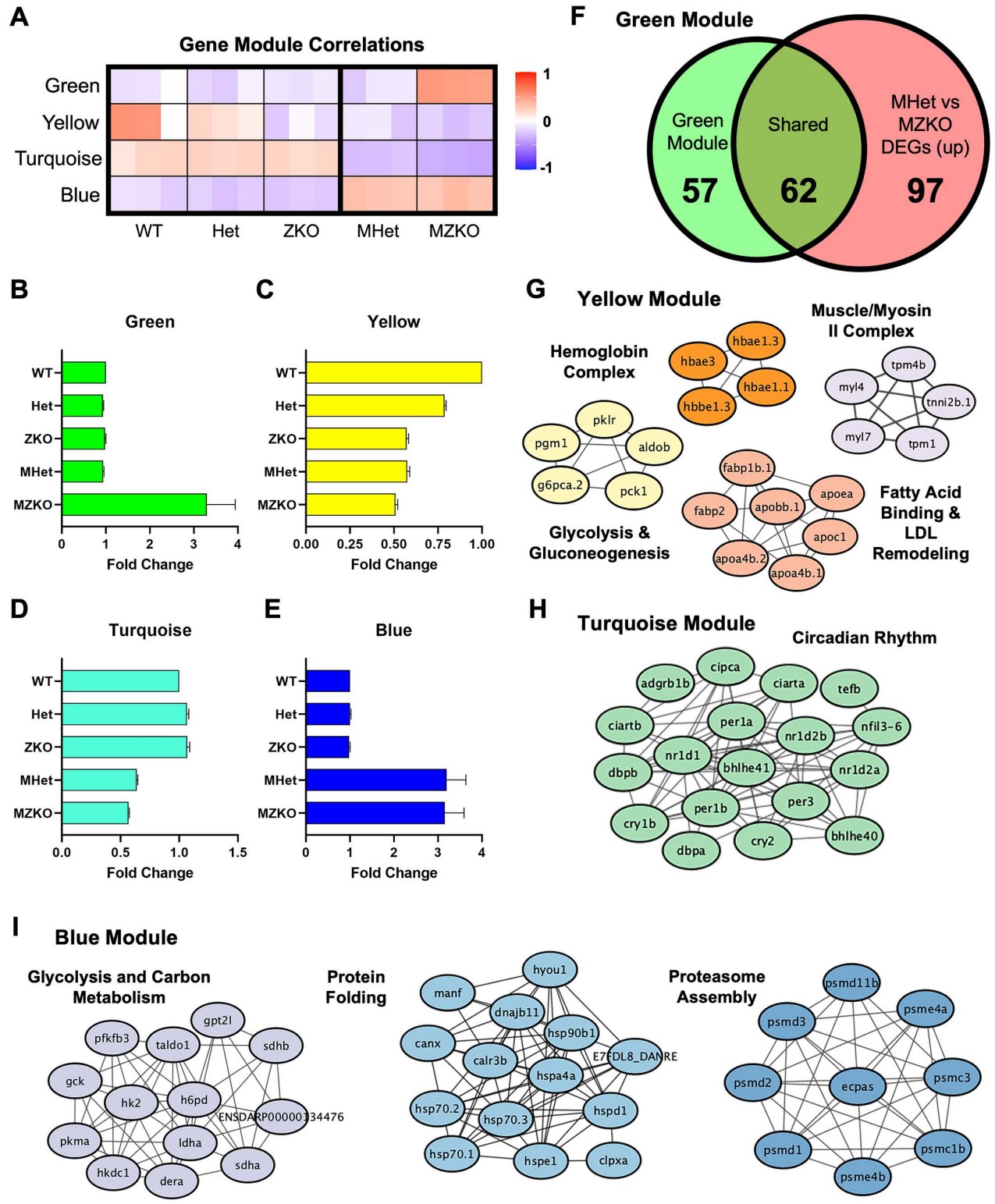

**Fig 8. Expression signatures derived from maternal and zygotic genotypes. A**: Heat map of modules identified through weighted gene co-expression network analysis that were highly correlated with either the maternal or zygotic genotype at 5 dpf. **B–E**: Fold change of normalized expression in each module across genotypes compared to the WT samples as validation of gene module correlations. **F**: Overlap of genes within the

green module and upregulated DEGs in MZKOs compared to their MHet siblings at 5 dpf shown in Fig 7C. **G**: STRING analysis of genes within the yellow module showing interacting networks of proteins involved in fatty acid binding and LDL remodeling, glycolysis and gluconeogenesis, the myosin II complex and the hemoglobin complex. **H**: STRING analysis of genes within the turquoise module showing an interacting network of proteins involved in circadian rhythm regulation. **I**: STRING analysis of genes within the blue module showing interacting networks of proteins involved in processes such as protein folding, carbon metabolism, and proteasome assembly.

indicating decreased expression (Fig 8B–8E). Surprisingly, four of the seven modules were driven by either disease state or maternal state.

The green module was strongly linked to disease progression, as these genes were positively correlated with the MZKO genotype, and many of these overlapped with the upregulated DEGs in MZKOs described above (Fig 8F). The yellow module, in contrast, suggested shared dysregulation due to loss of *pomgnt2*, whether maternal, zygotic, or both. It included 167 genes that were negatively correlated and downregulated across ZKOs and both genotypes obtained from KO x Het crosses. This module showed enrichment of smaller STRING networks involved in glycolysis and gluconeogenesis, the hemoglobin complex, and lipid metabolism and LDL remodeling (Fig 8G). The LDL network was driven by reduced *apolipoprotein* expression, similar to what was observed in MZKOs at 10 dpf indicating early dysregulation in lipid metabolism in ZKOs and their offspring. A few genes involved in muscle contraction were also present in the yellow module, including *myosin light chain 4 and 7* (*myl4, myl7*), *tropomyosin 1 and 4b* (*tpm1, tpm4b*), and *troponin I 2B* (*tnni2b.1*) (Fig 8G). This module suggested overall that certain differences in physiological processes may persist from ZKO females to their oocytes.

The turquoise and blue modules, which were respectively negatively and positively correlated with both MZKOs and their MHet siblings, revealed a difference between the offspring of Het females and ZKO females. Within the downregulated module (turquoise), numerous genes forming the structural constituents of different tissues were identified. These included multiple *crystallin* genes, which form the lens of the eye; *collagens*, the structural components of the ECM; *tubulins,* intermediate filaments, and genes involved in actin monomer binding that are critical for cytoskeletal assembly (S7A Fig). Interestingly, key regulators of circadian rhythm were also included in this module, including *per1a, per1b, per3, nr1d1, nr1d2a, nr1d2b, cry1b,* and *cry2* (Fig 8H). Within the entire turquoise module, *per1b* and *nr1d1* were the most centrally connected hub genes (S7B Fig), suggesting potential drastic differences in regulation of circadian rhythm stemming from the maternal state. The upregulated module (blue), in contrast, showed increased activation of several processes including protein folding with upregulation of heat shock proteins and other chaperones, proteasome assembly, and carbon metabolism, including several glycolytic enzymes (Fig 8I).

WGCNA provides gene network correlations, but to better understand these changes and their impact on the MHet and MZKO, we performed additional validation via qPCR using an array targeting 86 genes involved in glucose metabolism spanning glycolysis, the tricaboxylic acid (TCA) and pentose phosphate cycles, and gluconeogenesis (S3 File). By comparing 5 dpf WT, ZKO, MHet, and MZKO in the same array, we revealed a more complex pattern of up and downregulation consistent with broad glycolysis and TCA reduction in the MZKO with concurrent increase in gluconeogenesis (*gpca.1, fbp2*) (Figs 9A and S8). These changes included the hexokinase isoenzyme switch with glucokinase (*gck*) downregulation and upregulation of hexokinase 2 (*hk2*) found in metabolic reprogramming in cancer [47]. In parallel, there was upregulation of multiple genes in the pentose phosphate pathway leading to D-ribose production (Fig 9B). The MHet had similar changes in glycolysis that were less pronounced with limited involvement of the TCA cycle, but notably no *gck* downregulation (Figs 9A, 9B and S8). Serendipitously, the immune regulator β-2-microglobulin (*b2m*), which is essential for MCH class I function, was included among the positive controls and showed around 3-fold upregulation in both MHet and ZKO indicating immune activation (Fig 9C). Similar types of dysregulation were found in other pathways in the RNA-seq modules. For example, vitamin D activating enzyme transcripts (*cyp2r1* and *cyp27b1*) were down regulated in MHet and ZKO in the turquoise module, while the catabolic enzyme (*cyp24a1*) was upregulated (S2 File), showing birectional

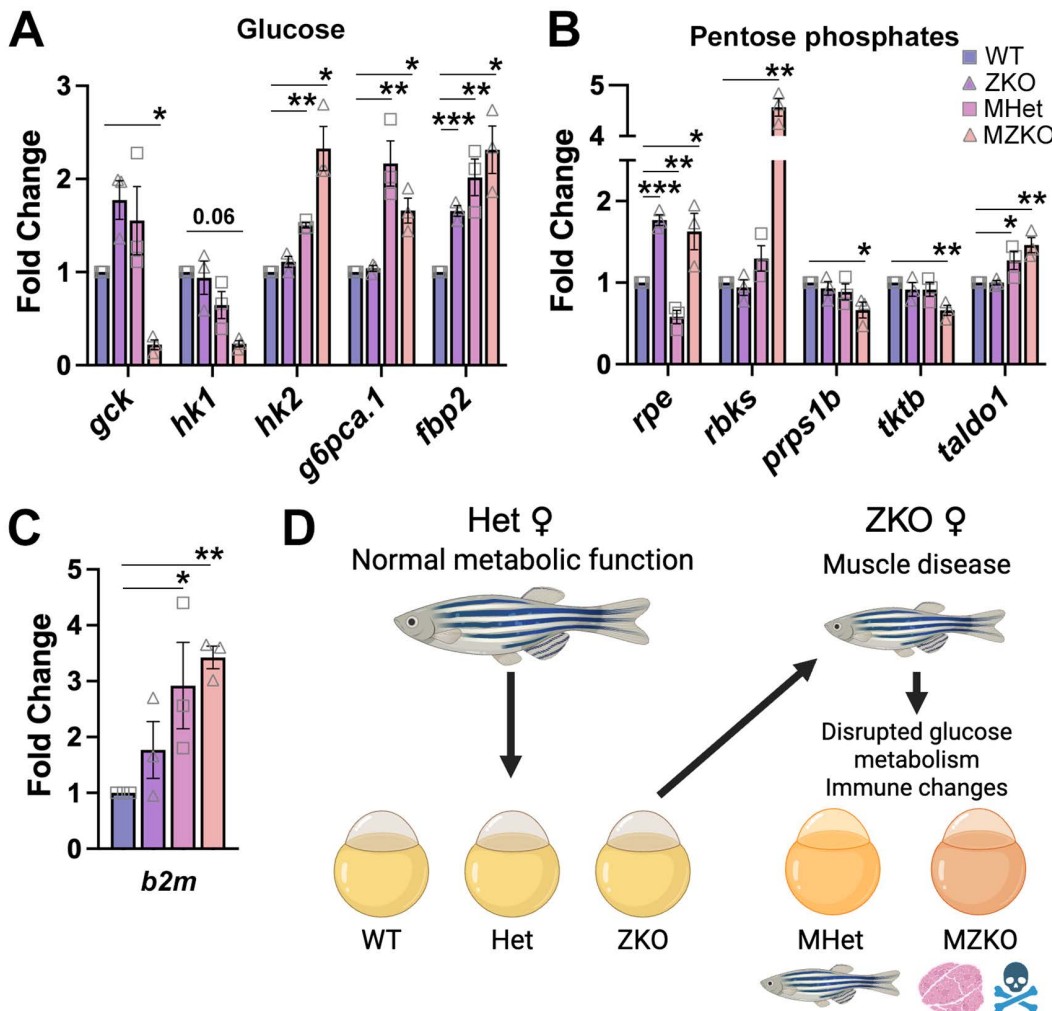

**Fig 9. Variable metabolic differences in progeny of Heterozygous and ZKO females. A.** Changes in glucose metabolism quantified via qPCR Qiagen RT² Profiler arrays show a switch between *gck* and *hk2* expression in glucose phosphorylation in MZKO and an increase in enzymes involved in glucone-ogenesis in both MHet and MZKO. **B.** Multiple changes in the pentose phosphate pathway are also noted mostly in MZKO. *rpe* which is involved in protection from oxidative stress is upregulated in both ZKO and MZKO but downregulated in MHet. **C.** *b2m* is upregulated in both MHet and MZKO. **D.** Model of maternal disruption affecting both MHet and MZKO at different levels. Created in BioRender. Manzini, M.C. (2026) https://BioRender.com/3rj87ca.

changes controlling different metabolic functions. A few changes were noted in the ZKO, but they were much more limited and did not reflect major disruptions in glucose metabolism (S3 File).

Overall, our findings show how removal of maternal *pomgnt2* mRNA in oocytes is necessary to unmask developmental phenotypes that recapitulate the core dystroglycanopathy phenotypes. However, our transcriptomic analyses revealed significant differences in multiple physiological processes correlated with maternal genotype that were more pronounced in the ZKO possibly reflecting disease onset. Based on these data, we propose a model in which metabolic dysfunction stemming from progressive muscle disease in ZKO females results in metabolic rewiring in the offspring (Fig 9D).

## Discussion

In this study, we characterized a novel zebrafish mutant for the glycosyltransferase Pomgnt2 to model a severe neuromuscular disorder caused by loss α-DG glycosylation, which mediates cell-ECM interactions that are critical for

morphogenesis and maintenance of multiple tissues [21,23,30,48]. We found that maternal *pomgnt2* leads to residual α-DG glycosylation in ZKO larvae that masked developmental phenotypes. By generating MZKOs from ZKO females, we achieved complete loss of α-DG glycosylation in the embryo. This recapitulated muscle, eye, and axon guidance deficits found in severe dystroglycanopathy and uncovered molecular signatures of disease progression and pathophysiology [32,33,36].

## Zebrafish models of dystroglycanopathy and maternal contributions

The zebrafish has become a valuable disease model for dystroglycanopathy as mouse KOs for dystroglycan (*Dag1*) and for the glycosyltransferases initiating O-mannosylation, *Pomt1* and *Pomt2*, show early embryonic lethality due a rodent-specific disruption in Reichert's membrane before placental formation [25–27]. To date, zebrafish KOs for *dag1*, *pomt1*, and *pomt2* are so far the only vertebrate animals to model global developmental loss of these genes. While *dag1* ZKOs show early lethality and progression of muscle disease consistent with morphants for other dystroglycanopathy genes [32], *pomt2* ZKOs displayed muscle, brain, and eye phenotypes only after 2 mpf [35]. Our studies on *pomt1* KOs reconciled these phenotypic difference by showing that later disease onset was due to maternally provided mRNA that could be removed by generating maternal-zygotic mutants [36].

The findings in this study confirmed that *pomgnt2* is also maternally provided, as ZKOs had delayed phenotypic onset while MZKOs are consistent with *pomt1* MZKOs, and with ZKOs for *dag1* and *fkrp* which are not maternally provided. This will likely extend to other zebrafish models of dystroglycan function. For example, *pomgnt1* ZKOs have only shown retinal degeneration at 6 mpf while mouse models have phenotypes consistent with dystroglycanopathy [34]. Transcripts for *pomt2*, *pomgnt1* and most other glycosyltransferases involved in α-DG glycosylation are detectable in zebrafish zygote before the MZT, strongly suggesting a maternal effect [37].

One striking observation in this study and our past work on *pomt1* is that ZKOs retain α-DG glycosylation for at least 5–7 dpf indicating that the proteins may still be functional for days beyond the MZT [49]. Prior studies in mouse tissue have shown that glycosylated α-DG has a half-life of approximately 3 weeks [50], which may offer a partial explanation for the prolonged stabilization of glycosylation in *pomgnt2* and *pomt1* ZKOs. However, our past work also showed that the Pomt1 protein was present in *pomt1* ZKOs at 5 dpf [36]. O-glycosyltransferases *pomt1* and *pomgnt2* join the class of maternally provided transcripts shaping larval morphogenesis after the MZT. While the best characterized maternal transcripts are involved in the coordination of zygotic genome activation and embryonic polarity in fast-developing embryos like the zebrafish [47], multiple maternal zygotic transcripts are involved in later developmental process such as skeletal formation and neuronal differentiation [15,16,51]. How these transcripts remain active at later stages of differentiation is still unknown. Transcriptomic and proteomic profiling in oocytes from another teleost, the pikeperch (*Sander lucioperca*), suggested that the mother can provide RNA binding proteins to stabilize transcripts involved in neurogenesis and metabolism [47]. Recent single-cell transcriptomic analysis with metabolic labeling during the MZT supported this hypothesis of differential maternal transcript stability dependent on cell type [47]. While there are no suitable antibodies to detect Pomgnt2 in zebrafish, this warrants further investigation into potential mechanisms to stabilize maternal transcripts and their proteins during and after the MZT.

## Lipid metabolism disruptions due to *pomgnt2* loss of function

In mice, cortical migration deficits caused by loss of *Pomgnt2* have been well-documented and resemble cobblestone lissencephaly observed in patients with WWS. However, *Pomgnt2* KO pups die within the first postnatal day, and characterization of muscle and eye deficits has not been conducted in a global KO model [30,31]. In addition, little is known about disease progression in *POMGNT2*-related dystroglycanopathy due to the limited number of reported cases.

While the generation of *pomgnt2* MZKOs unmasked early disease phenotypes compared to ZKOs, the presentation especially in the brain was milder than previously characterized *pomgnt2* morphants which presented with developmental

brain and eye defects by 48 hpf [23]. Differences between knockdown strategies (MO-mediated or RNA interference) have been often reported in cell lines and mouse and zebrafish models, and milder phenotypes have also been reported in CRISPR/Cas9-edited organisms [52]. MO-mediated knockdown strategies have higher potential for off-target effects especially affecting the brain compared to stable KO lines [53]. Previous work on *pomgnt2* morphants also showed that while the phenotype could be improved by WT mRNA injection, it was never completely rescued [23]. Genetic zebrafish models generated through various approaches (*dag1* ZKO – ENU, *fkrp* ZKO – TALEN, *pomt1* MZKO – ENU, *pomgnt2* MZKO – CRISPR/Cas9) show generally consistent disease progression, but limited brain involvement, which could be due to fundamental evolutionary differences including the absence of processes involved in cortical migration and lamination that are disrupted in higher vertebrates [32,33,36]. For this reason, we have primarily focused on highly conserved axon guidance mechanisms controlling retinal ganglion neuron pathfinding from the retina to the tectum, where findings in dystroglycanopathy mouse models can be replicated.

Our developmental analysis of muscle integrity showed the most consistent findings. *pomgnt2* MZKOs show progressive detachment of muscle fibers from the MTJs and degeneration of the slow-twitch fiber layer as muscle integrity is lost, similar to DMD (*dmd-sapje*) and LAMA2-related dystrophy (*lama2-candyfloss*) zebrafish models [38]. Similar phenotypes have been also noted in a morphant model for *fkrp* [39]. The rapid fiber degeneration in our model is more similar to DMD models, while detached fibers in the *lama2* zebrafish KO can survive for days following detachment [54], suggesting shared pathophysiology among mutants in the DGC. Transcriptomic analysis during severe disease at 10 dpf showed widespread reduction in muscle contraction gene expression, along with alterations in genes controlling lipid metabolism and sterol biosynthesis that could be secondary to muscle disease and sarcolemma breakdown.

Dyslipidemia and other disruptions in lipid metabolism are well-characterized in DMD [55–57]. Studies on individuals with DMD and animal models not only show different profiles of cholesterol, phospholipids, and fatty acids in dystrophic muscle and serum, but also identified changes in membrane phospholipid composition and fatty acid metabolism in mitochondria that actively contribute to muscle disease progression [58–60]. Much less is known about these processes in dystroglycanopathies. One metabolomic study in mice using a knock-in *Fkrp* line (*Fkrp$^{P448L}$*) to model a less severe form of dystroglycanopathy, Limb Girdle Muscular Dystrophy 2i (LGMD2i), identified global metabolic perturbations with increases in glycolytic intermediates and lipid metabolites [61]. Our findings in 10 dpf MZKOs demonstrate broad downregulation of genes involved in cholesterol, phosphoglyceride, and lipoprotein metabolism, again suggesting shared pathophysiology with DMD that warrants further investigation.

## Metabolic impact of having a knock-out mother

In addition to defining differences in disease progression and severity in *pomgnt2* ZKOs and MZKOs, our WGCNA analyses revealed clear transcriptomic differences associated with maternal genotype, highlighting important consideration specific to generating maternal-zygotic KOs through zygotic KO females. Offspring of ZKO females showed decreased expression of structural genes (collagens, tubulins, and crystallins), suggesting potential differences in tissue formation and development. In addition, the reduction in circadian regulators warrants further investigation into possible differences in rhythmicity. In contrast, gene expression changes altering protein homeostasis and glycolysis suggested that offspring of ZKO mothers may be primed for metabolic disruptions. Detailed analysis of glucose metabolism changes identified moderate reductions in glycolysis, TCA cycle, and glycogen metabolism in MZKOs at 5 dpf that matched more severe changes found at 10 dpf with a subset of the same genes also altered in MHets. This was paired with an increase in the MHC class I component *b2m* which is an established biomarker of immune activation leading to worse prognosis in cancer and multiple diseases [62–65]. Despite these gene expression differences, MHets are behaviorally and histologically indistinguishable from WT and Het larvae from HetxHet crosses. Expression changes appear overall more modest in MHets and may not have major deleterious impact on the overall health of the offspring, but could modify disease progression in MZKOs. Fewer metabolic gene expression changes were noted in ZKOs as early as 5 dpf when they still benefit

from residual α-DG glycosylation. It is possible that ZKOs have life-long changes that compound with progressive muscle disease in adult females with consequences for the maternal nutrition and metabolism altering the yolk of the oocytes, which is rich in lipids and proteins [66,67]. Several studies have also noted reduced offspring viability and other abnormalities in zebrafish in response to metabolic disruptions in the female parents [55,68,69]. These differences could be leveraged in future studies to identify novel disease modifiers.

In conclusion, our study shows how a maternal-zygotic mutant can be leveraged to unveil early disease phenotypes in strains impacted by maternal compensation, while highlighting physiological differences that emerge when offspring are obtained from zygotic mutant females. These findings have immediate relevance for other zebrafish mutants of dystroglycan-related disorders as well as other zebrafish mutants where maternal compensation is suspected.

## Materials and methods

### Ethics statement

All experiments and procedures involving live animals in this study were approved by the Institutional Animal Care and Use committee of Rutgers University under protocol PROTO201900047.

### Experimental model

Zebrafish were housed on a recirculating Tecniplast USA system under a 14–10 light-dark cycle at 28°C in 3.5 L tanks and fed twice daily. For spawning events, males and females were placed off-system in divided spawning cages each evening, and the dividers were removed the following morning following the start of the lights on period. Embryos obtained from spawning events were collected in petri dishes with egg water containing methylene blue ($5x10^{-5}$% w/v) until 5–6 days post fertilization (dpf) when they were placed on the system for raising or until their respective endpoints.

### Guide RNA (gRNA) assembly

Three guide RNA (gRNA) target sequences were identified from the Burgess Lab UCSC Track Data Hub for CRISPR targets and ordered from Integrated DNA Technologies (IDT; Coralville, IA) with SP6 or T7 promoter sequence, gRNA target sequence, and TracrRNA overlap sequencing. 10 µM of each gRNA oligo and 10 µM of universal oligo were annealed in 25 µl reactions with 1U Phusion High Fidelity DNA Polymerase (Thermo Fisher Scientific) in a T100 Thermal Cycler (Bio-Rad) under the following cycling conditions: 98°C for 2 minutes, 50°C for 10 minutes, and 72°C for 10 minutes. gRNAs were then synthesized using a HiScribe SP6 or T7 Quick High Yield RNA Synthesis kit (New England Biolabs) in a 30 µl containing 3 µl of annealed oligo product at 37°C overnight. gRNAs were then purified using an RNA Clean & Concentrator kit (Zymo Research), diluted in sterile, nuclease free water, and stored at -80°C for no longer than one month.

### Generation of the *pomgnt2* line

Microinjections were performed on one cell stage EK zebrafish embryos using 1 nanoliter/embryo of injection master mix containing gRNA and recombinant Cas9 protein (PNA Bio Inc). Injections using a gRNA targeting *tyrosinase* (*tyr*) was used to determine technique efficiency for each experiment by screening the developing embryos for loss of pigment. Uninjected clutchmate controls were also used in every experiment. The injected embryos were housed in embryo medium with methylene blue and screened periodically for 5 days to remove dead and deformed embryos, and the remaining were placed on the system to be raised for mutant line propagation. Between 3–4 months post fertilization (mpf), surviving injected zebrafish were anesthetized in 0.016% w/v tricaine methane sulfonate (Tricaine, MS-222) and fin clipped. DNA was extracted and heteroduplex mobility assays were used to identify fish harboring indels successfully induced through CRISPR-Cas9. The fish demonstrating the highest degree of heteroduplex formation were then outcrossed with WT EK zebrafish to generate $F_1$ founders.

## F$_1$ founder selection

When potential F$_1$ founders reached 3–4 months of age, they were anesthetized and fin clipped, and DNA was extracted and amplified from fin clips as previously done for the F$_0$ generation. PCR products and forward primers for each reaction were sent to Azenta Life Sciences (South Plainfield, NJ) for Sanger Sequencing. Five F$_1$ founders (4 female, 1 male) with a 13 bp insertion and 4 bp deletion in exon 1 along with a 7 bp deletion in exon 2 were selected to propagate the main mutant line described in this study. Additional founders with only the exon 1 mutations or only the exon 2 mutations were identified and later used to corroborate loss of pomgnt2 function. F$_1$ founders were then outcrossed again to WT EK fish to further propagate the line.

## Genotyping of the *pomgnt2* line

Following the identification of the primary set of F$_1$ founder mutations, we used several validated methods to genotype the fish for experiments. For the HetxHet crosses, DNA was extracted and amplified exactly as done for the F$_0$ and F$_1$ generation. Following PCR amplification, 10 μl of PCR product was digested at 37°C for 1–2 hours to overnight with 1U SnaBI restriction enzyme, which recognizes the sequence that is deleted in exon 2 (New England Biolabs) in a 25 μl reaction. Digestion was then stopped by incubating the samples at 4°C. Each sample was run on a 1.5% agarose gel at 90 V for 1 hour and imaged under UV light to visualize banding pattern. Random samples were also periodically spot checked via Sanger Sequencing using amplicons of both the exon 1 and 2 mutations to further ensure genotyping accuracy. Genotyping was also performed through quantitative allele-specific PCR with custom designed Affinity Plus qPCR probes and primers from Integrated DNA Technologies (Coralville, IA) on a QuantStudio 6 qPCR system (Applied Biosystems) for 40 cycles with an annealing temperature of 62°C. Due to differences in WT and mutant probe efficiency, qPCR-based genotyping reactions were often run with a second reaction containing only the WT probe, along with multiple controls genotyped through alternate methods (i.e., restriction digest or Sanger sequencing) to ensure genotypic accuracy.

## Survival and morphological analysis

Progeny of Heterozygous (HetxHet) crosses were placed on system at 5 dpf ungenotyped with reduced water flow as close to maximum allowable stocking densities as possible (5–30 dpf: 14/L; > 30 dpf: 4/L) to induce competition for food. The same process was performed for KOxHet survival and morphological analyses with a slightly reduced initial stocking density to accommodate smaller clutches generated from KO females (11–13/L). At regularly schedule intervals of 1 month, 3 months, and 5 months for HetxHet crosses and 7, 10, 14, and 28 days for KOxHet crosses, the animals were removed from the system, imaged to obtain body length measurements, and either fin clipped and returned to the system for breeding and other experimental purposes, or sacrificed for other experimental purposes. At timepoints of 1 month of less, the juveniles were imaged in 3% methylcellulose with a M165 FC stereo microscope and LAS Software v4.21 (Leica Microsystems). At 3 and 5 months, the fish were imaged with a handheld camera to obtain body length measurements. Body length measurements were taken from the mouth to the base of the tail fin in ImageJ with the researcher masked to genotype.

## RNA isolation

Total RNA was extracted with RNAzol RT RNA Isolation reagent as follows: Each sample was homogenized with 500 μl of RNAzol. 200 μl of nuclease free water was added to each sample, which was then mixed vigorously for 15 seconds and incubated at room temperature for 15 minutes. The samples were then centrifuged at 12,000 x g for 15 minutes at 4°C. 600 μl of supernatant was then removed and added to a fresh 1.5 ml Eppendorf tube with 600 μl of 100% isopropanol and mixed by inverting. Samples were then incubated at -80°C for 30 minutes, then at room temperature for 15 minutes, and then centrifuged at 12,000 x g for 10 minutes at 4°C. The supernatant was then removed, and the pellet was washed with

1 ml of 75% ethanol and centrifuged at 8,000 x g for 3 minutes at 4°C. The supernatant was removed, and the pellet was washed in 100 µl of 75% ethanol 3–4 more times with centrifuging in between. The pellet was then dried on a heat block at 50°C, resuspended in nuclease free water, and stored at -80°C for used in downstream applications.

## Quantitative reverse transcriptase PCR (qRT-PCR)

qRT-PCR was performed to evaluate *pomgnt2* gene expression across a time course in HetxHet crosses at 20, 40, 60, and 90 dpf in composite samples of genotyped heads (20, 40, and 60 dpf) or tails (90 dpf). Following RNA isolation, 1 µg of RNA was reverse transcribed using iScript Reverse Transcription (RT) Supermix (Bio-Rad) in a 20 µl reaction containing 16 µl of RNA and 4 µl of RT. cDNA was then diluted from 50 ng/µl to 20 ng/µl in nuclease free water. qPCR reactions were run using PowerUp SYBR Green Master Mix (Applied Biosystems) in triplicate 15 µl reactions with final cDNA and primer concentrations of 10 ng/ul and 400 nM, respectively, on a QuantStudio 3 qPCR system (Applied Biosystems) for 40 cycles with annealing temperatures of 56°C for the *pomgnt2* and 53°C for *rpl13α*, the endogenous control. Each experiment was run with no template controls and 1–2 no RT controls per primer pair. Each datapoint is presented as fold change ($2^{-\Delta\Delta CT}$) compared to the average of the WT samples.

## Western blotting and WGA enrichment

The western blotting procedure used in this study was previously described by Karas et al. [36]. Western blotting was performed on composite samples of 7–10 30 dpf fish using wheat germ agglutinin (WGA) enrichment of glycoproteins. Composite samples were lysed in 100 µl of buffer made in-house (50 mM Tris pH 8, 100 mM NaCl, 1 mM PMSF, 1 mM Na Orthovanadate, 1% Triton-X 100), centrifuged at 4°C for 30 minutes, sonicated 3 times for 5 minutes, and treated with 2 µl DNase I (New England Biolabs). Protein concentration in each lysate was quantified using a BCA Protein Assay kit (G-Biosciences). 500 µg of protein was diluted in 200 µl of lectin binding buffer (20 mM Tris pH 8, 1 mM $MnCl_2$ and 1 mM $CaCl_2$), added to 50 µl of WGA agarose-bound beads (Vector Laboratories), and incubated overnight, rocking at 4°C. The following morning, the samples were centrifuged at 15,000 x g for 2 minutes to remove the supernatant, and the bead-bound glycoproteins were eluted with 30 µl of 4X Laemmli buffer (Bio-Rad). Glycoproteins were separated on a 4–12% Bis-Tris Protein SDS-PAGE gel (Life Technologies) run at 55 V for 10 minutes, then 120 V for 70 minutes, in 1X Invitrogen NuPAGE MOPS SDS Running Buffer. The gel was then incubated in transfer buffer (70% Water, 20% Methanol, 10% 1X Tris/Glycine Buffer with 0.1% SDS (Bio-Rad) for 5–10 minutes. Proteins were then transferred to a nitrocellulose membrane (Thermo Fisher) at 4°C in transfer buffer with an ice pack at 110 V for 3.5 hours. The membranes were then stained with 0.1% naphthol blue black (amido black) (Millipore Sigma) for total protein staining. Nitrocellulose membranes were blocked in 5% milk diluted in 1X TBS containing 0.1% Tween-20 (TBS-T), then probed with 1:100 anti-α-dystroglycan antibody clone IIH6C4 (Millipore Sigma) and 1:1000 anti-*β*-dystroglycan antibody (ab62373, Abcam) overnight at 4°C. The following day, the membranes were briefly rinsed in MilliQ water, probed with 1:3000 peroxidase AffiniPure donkey anti-mouse IgG and 1:20 000 peroxidase AffiniPure donkey anti-rabbit IgG (Jackson ImmunoResearch), washed in 1X TBS-0.1% Tween-20 5X for 5 minutes, and developed with chemiluminescence Pierce ECL Western Blotting Substrate on CL-XPosure Film (Thermo Fisher).

## Automated behavior tracking in larvae

Automated tracking of locomotor behavior in larvae was performed using a DanioVision Observation Chamber (Noldus Information Technology, Wageningen) and EthoVision XT Video Tracking software at 30 frames per second. All experiments were done in a randomized, ungenotyped manner. At 5 dpf, larvae were placed in a 96 well polystyrene cell culture plate in equal water volumes and placed in the observation chamber for a total of one hour: the first 30 minutes for habituation, and the last 30 minutes for tracking. At 7, 10, and 14 dpf, larvae were placed in a 24 well polystyrene cell culture plate to increase the total area for movement. These experiments were performed in the same manner, but with 20

minutes of habituation and 20 minutes of tracking. The observation chamber was always held at a constant 28.5°C and experiments always began at the same time each day to minimize variability.

## Behavior tracking in adult fish

Adult WT and ZKO fish were evaluated for locomotor function at approximately one year of age. Each fish had been genotyped prior to the experiment, which occurred over the course of four days, evaluating one fish at a time in alternating WT-ZKO order whenever possible. The experiments always began at the same time each day to minimize variability between experiment days. Each fish was placed in an arena of 60 cm in diameter held at approximately 26°C ± 1.5. The fish were habituated for 5 minutes, followed by 10 minutes of recording using a GS3-U3-41C6NIR-C 1" FLIR Grasshopper video camera at 30 frames per second. Swimming trajectories were extracted from the video data using ZebraZoom software [70].

## Cryosectioning and fluorescent immunohistochemistry

All fish, larvae, juvenile, or adult, were fixed in 4% paraformaldehyde (PFA) overnight at 4°C. The fish were in 1X PBS 3 times for 5 minutes to remove any residual PFA. The fish were then transferred to 15% sucrose with 0.2% sodium azide in 1X PBS for 1–2 days, followed by 30% sucrose with 0.2% sodium azide in 1X PBS for 1–2 days, and then frozen in isopentane on dry ice in Tissue Freezing Medium (Ted Pella Inc) and stored at -80°C. Fixed tissue from adult fish was also decalcified for 1–2 hours in Cal-Ex (Fisher Scientific) and washed in 1X PBS 3 times for 5 minutes before beginning cryoprotection in sucrose. Cryosectioning was performed on a Leica CM1850 UV Cryostat (Leica Microsystems) generating 12 μm transverse sections of the eyes and brain in larvae and 16 μm transverse sections of muscle in adults. The sections were dried on a slide warmer and either stained immediately or stored at -20°C until staining.

For transverse sections of adult muscle, sections were outlined, blocked in 10% normal goat serum (NGS) containing 1% Triton X 100 and 2% Tween-20 in a humidified slide box at room temperature for 1 hour. Next, the sections were incubated in primary antibody diluted in 1% NGS containing 1% Triton X 100 and 2% Tween-20 overnight at 4°C. The primary antibodies used in these experiments were 1:50 anti-laminin L9393 (Millipore Sigma) and 1:50 anti-DAG1 antibody clone IIH6C4 (Abcam). The following day, the sections were washed in 1X PBS 3 times for 5 minutes and incubated in secondary antibody diluted in 1% NGS containing 1% Triton X 100 and 2% Tween-20 at room temperature for one hour. Alexa Fluor goat anti-rabbit and goat anti-mouse secondaries (Thermo Fisher) were used in these experiments at 1:250 concentration. Sections were then counterstained in 1X DAPI or Hoechst for 10 minutes, coverslipped with VWR micro cover glasses in Prolong Gold Antifade Mountant (Thermo Fisher) and cured for at least 24 hours before imaging.

For transverse sections of the eyes and brain in larvae, this process was repeated with the following modifications: blocking was performed in 10% NGS with 1% Triton X 100, primary and secondary antibodies were diluted in 1% NGS with 0.1% Triton X 100, and counterstaining in 1X DAPI was done for 1 hour. The primary antibodies used in these experiments were 1:200 anti-synaptophysin (Abcam), 1:200 zpr1 (ZIRC), and 1:100 zn-8 (Developmental Studies Hybridoma Bank). The secondary antibodies and dilutions were unchanged. All cryosections were imaged on a Zeiss LSM800 confocal microscope with Zeiss Zen imaging software. Objectives used were all Zeiss Plan-APOCHROMAT: air immersion 10X (NA = 0.45) and 20X (NA = 0.8), and oil-immersion 40X (NA = 1.4) and 63X (NA = 1.4).

## Whole mount fluorescent immunohistochemistry

The protocol for whole mount staining was adapted from Bailey et al. [71]. Fluorescent immunohistochemistry to analyze muscle in larvae was performed through a whole mount staining protocol adapted from Bailey et al. Zebrafish fixed overnight in 4% PFA at 4°C. The following day, they were rinsed 3 times for 10 minutes in 1X PBS-0.1% Tween 20 (PBS-T), followed by permeabilization in 1 mg/ml collagenase D (Sigma) for 1.5 hours at room temperature. The fish were then washed 3 times for 10 minutes in PBS-T. For experiments using Alexa Fluor Phalloidin 546 (Thermo Fisher) and/or Alexa

Fluor 488 α-bungarotoxin conjugate (Invitrogen), these steps were performed at this stage at 1:20 and 1:500 dilutions, respectively, for 2 hours at room temperature, followed by additional washes in PBS-T. The fish were then blocked overnight in antibody blocking solution (Ab block) made in-house (5% BSA, 1% DMSO, 1% Triton-X-100, 0.2% saponin in 1X PBS) at 4°C. The following day, the fish were moved to primary antibody diluted in Ab block. The primary antibodies used in these experiments were 1:50 anti-DAG1 antibody clone IIH6C4 (ab234587, Abcam), 1:50 anti-laminin L9393 (Millipore Sigma), 1:25 F59 (ZIRC), and 1:10 anti-SV2 (Developmental Studies Hybridoma Bank). Primary antibody incubations were performed overnight at 4°C. The following day, the fish were then washed out of primary in PBS-T 3 times for 10 minutes each and moved to secondary antibody diluted in Ab block. 1:250 Alexa Fluor goat anti-mouse and goat anti-rabbit secondaries (Thermo Fisher) were used for these experiments. The following day, the fish were washed out of secondary in PBS-T, mounted in 1% low melt agarose, and imaged on a Zeiss LSM800 confocal microscope with Zeiss Zen imaging software.

### RNA-sequencing and analysis

RNA sequencing was performed by Novogene Corporation (Sacramento, CA) on RNA extracted from composite samples of 4–6 whole larvae at 5 and 10 dpf. At 4 dpf, DNA was extracted from live progeny of HetxHet and KOxHet crosses using a Zebrafish Embryonic Genotyper (ZEG) Microfluidic system (wFluidx Inc) and genotyped directly using our custom qPCR assay. At 5 dpf, the fish were sorted by genotype, euthanized in Tricaine, and snap frozen in liquid nitrogen. Fish used for the 10 dpf experiment were housed on system separated by genotype until this timepoint. Total RNA was extracted using RNAzol RT RNA Isolation reagent as described in the **RNA isolation** section and shipped overnight on dry ice for sequencing. In addition, leftover RNA from the composite samples was reverse transcribed using iScript Reverse Transcription Supermix (Bio-Rad) as described in the "Quantitative reverse transcriptase PCR (qRT-PCR)" section and genotyped again using our Affinity Plus qPCR probes to ensure all fish were sorted correctly.

143.6 Gb of raw data were delivered as fastq files with an average of 44.5 million raw reads for the 5 dpf experiment and 48.4 million reads for the 10 dpf experiment, which were uploaded to Amarel, the high-performance computing cluster of Rutgers University. Quality control was performed in house using FastQC [72]. Following FASTQC warnings in Per Base Sequence Content, the first 10 bp of each read was trimmed to remove residual adapter sequence bias and reads with Phred scores <20 were filtered out before alignment. Filtered sequencing reads were aligned to the GRCz11 (danRer11) genome build with HISAT2 [73] followed by sorting and indexing through Samtools [74], each of which were performed using default settings. Filtered sequencing reads were aligned to the GRCz11 (danRer11) genome build with HISAT2 [73] followed by sorting and indexing through Samtools [74]. Alignment statistics for each sample are detailed in S2 Table. Mapped reads were counted from sorted bam files with the featurecounts command of the Subread package in R (v4.4.1) [75] using the Lawson Lab Zebrafish Transcriptome Annotation v4.3.2 [76] as an index to improve mapping through more comprehensively defined 3'UTR annotations. Count matrices were then normalized through DESeq2 [77] and differential expression was calculated across genotypes with a adjusted p-value <0.01 and fold change >0.7 or <-0.7 to determine significance. Enrichment analysis was performed in STRING through manual curation using a combination of available databases including Gene Ontology, Zebrafish Phenotype Ontology, STRING Local Networks, COMPARTMENTS, and Pfam and Interpro Protein Domains. For raw counts and differential expression analyses, see S1 File.

### Weighted gene co-expression network analysis

Following differential expression analysis, weighted gene co-expression network analysis (WGCNA) [78,79] was performed in R (v4.4.1) on 5 dpf RNA sequencing reads in WT, Het, ZKO, MHet, and MZKO samples. This timepoint was selected to perform a direct comparison between HetxHet and KOxHet progeny at a timepoint when residual α-dystroglycan glycosylation was still present in MZKOs. The data was variance stabilized and reduced to the 95th quantile leaving a total of 1818 genes remaining. Clustering was performed using Scale Free Topology Model Fit with a soft

thresholding power of 12. The genes were then clustered into modules of correlated genes which were prioritized based on correlations with both the female parent's genotype and the progeny's genotype. Genes belonging to each module are available in S2 File. To construct the networks for each module, edge lists were generated from topological overlap matrices (TOMs) using a minimum correlation threshold of 0.4. The top 50 edges were filtered and visualized in Cytoscape [80] to identify the most centrally connected hub genes for each module.

## Analysis of glucose metabolism via qPCR arrays

Expression of 84 genes involved in glucose metabolism was performed using Qiagen RT$^2$ Profiler PCR Arrays (PAZF-006Z) in the 384-well (E) configuration so that 4 independent samples can be tested in the same arrays. RNA was isolated using an RNAeasy kit (Qiagen) from pools of 10 5 dpf *pomgnt2* WT, ZKO, MHet and MZKO larvae. cDNA transcription was conducted on 400ng of mRNA per array using the RT$^2$ First Strand kit (Qiagen) which also includes genome elimination. qPCR was performed on a QuantStudio 6 system (Applied Biosystems), using RT$^2$ SYBR Green ROX Mastermix (Qiagen) following the manufacturer's guidelines for array analysis. Three independent arrays were run for all four samples. Data analysis was completed online at the GeneGlobe Analysis Center (https://geneglobe.qiagen.com/us/analyze). Results were normalized using the arithmetic mean of control genes (*acta1b*, *hprt1*, *nono*, and *rpl13a*), excluding *b2m* which showed a differential expression pattern in the MHet and MZKO. Average CT values and standard deviation, and Fold Change results are available in S3 File.

## Quantitative analysis of neuromuscular junctions

Maximum intensity projections of the NMJs, marked by SV2 and α-bungarotoxin (α-BTX), were generated from z-stacks taken with 1 μm intervals at 20X magnification. Each individual slice was scanned four times and averaged. The images were processed in Fiji and imported into a custom CellProfiler [81] pipeline. 3–4 hemi-segments and myotendinous junctions (MTJs) were traced per fish. The SV2 channel was rescaled identically across all samples when analyzing muscle fibers within the hemi-segments to minimize background fluorescence, but this was not necessary when analyzing MTJs. α-BTX and SV2 puncta were detected using Otsu's thresholding method with three classes. Pixel intensity was recorded for each puncta and averaged across each fish, and colocalization was quantified using Pearson's correlation coefficient. All NMJ images were acquired and analyzed with the researcher masked to genotype.

## Quantitative analysis of photoreceptor synapses

Raw czi files from cryosections of the entire retina were taken at 20X magnification with identical imaging parameters. The outer plexiform layer was traced in CellProfiler from the dorsal ciliary margin to the optic nerve exit without rescaling or any other modification to the image, and average pixel intensity was obtained. The images were analyzed with the researcher masked to genotype.

## Statistical analysis

All statistical analyses, aside from RNA sequencing analyses, were performed in GraphPad Prism v.8.20 (GraphPad, San Diego, CA). Normality of each dataset was assessed using a Shapiro-Wilk test. Statistical outliers were identified and removed from quantitative data sets using the ROUT method [82]. Outlier removal did not create novel significance for any experiment. For comparisons between 2 groups of a single measure, an unpaired t-test was used, or alternatively a Mann-Whitney test for non-normal datasets. A two-way ANOVA was used for comparisons between 2 or more groups across multiple conditions (i.e., timepoint). Statistical significance was defined as a p-value <0.05 (*<0.05; **<0.01; ***<0.001). For all experiments except for survival analyses, each datapoint represents an individual fish (n). For survival analyses, each datapoint represents the percent genotype of an independent clutch of fish (N), while the total number of individual fish (n) per clutch is listed in each figure legend. All error bars represent standard error of the mean (SEM).

## Supporting information

**S1 Fig. Additional validation of the pomgnt2 line. A**: Quantitative real-time PCR analysis of *pomgnt2* gene expression showing no significant differences in WTs, Hets, and ZKOs. **B**, **C**: Western blot analysis of glycoprotein enriched lysate showing that ZKOs with only the exon 1 (**B**) or exon 2 (**C**) mutations also show complete loss of α-DG glycosylation. **D**: Staining of muscle at 1 mpf with fluorescently conjugated phalloidin showing normal muscle fiber integrity in ZKOs (Scale Bar: 100 μm).
(TIF)

**S2 Fig. Muscle and motor phenotypes in 1 year old ZKOs. A**: Full transverse cryosections showing complete deterioration of muscle integrity in ZKOs reflective of advanced muscle disease, in addition to disrupted laminin staining around myofibers and increased nuclear staining (DAPI) suggestive of severe fibrosis (Scale Bars: 500 μm for whole cryosection, 20 μm for zoomed-in images of muscle). **B–D**: Comprehensive analysis of swimming behavior and locomotor function showing that ZKOs have reductions in standard measures such as distance (**B**), velocity (**C**), and acceleration (**D**).
(TIF)

**S3 Fig. Additional assessment of locomotor function in MZKOs. A**, **B**: Assessment of maximum velocity (**A**) and maximum acceleration (**B**) at 5 dpf showing significant reductions in the MZKOs. **C, D**: Assessment of total distance (**C**) and average velocity (**D**) showing significant reductions in the MZKOs at 7, 10, and 14 dpf.
(TIF)

**S4 Fig. *A*nalysis of horizontal cell processes in the outer plexiform layer (OPL) of pomgnt2 maternal mutants.** Retina sections from 10 dpf *pomgnt2* MHet and MZKO fish showed no difference in calbindin immunostaining outlining horizontal cell processes and a subset of bipolar cells. (63X magnification, scale bar: 10μm).
(TIF)

**S5 Fig. Evaluation of genetic compensation by dystroglycan and matriglycan-modifying enzymes. A, B**: FPKM values with unadjusted and adjusted p-values derived from RNA-seq experiments from HetxHet crosses (**A**) and KOxHet crosses (**B**).
(TIF)

**S6 Fig. Weighted gene co-expression network analysis pipeline and validation. A**: Data set normalization through DESeq2 and reduction to include genes expressed in the 95th quantile and above in all samples. **B**: Selection of soft thresholding power based on scale independence and mean connectivity. A soft thresholding value of 12 was selected. **C**: Dendrogram of gene modules derived from the reduced dataset of 1818 genes. The largest gene modules are turquoise and blue. **D**: Heat map of all gene modules identified, including those strongly correlated with maternal of zygotic genotype (green, yellow, turquoise, blue), and those that are not (brown, red, grey).
(TIF)

**S7 Fig. Additional characterization of the turquoise module. A**: Enrichment of crystallin, collagen, tubulin, and intermediate filament protein domains. **B**: Expression correlation analysis of the turquoise module revealing *per1b* and *nr1d1* to be the most centrally connected hub genes.
(TIF)

**S8 Fig. KEGG pathway representation of glycolysis/gluconeogenesis and TCA cycle changes in MHet and MZKO.** Transcripts tested in the arrays were mapped using KEGG Mapper (https://www.genome.jp/kegg/mapper/color.html) with respective KEGG identifiers and labeled in yellow when unchanged, red when increased and green when decreased. Outputs were modified by adding gene identifiers to ease identification. **A.** Glycolysis was overall reduced in MZKO with an increase in transcripts involved in gluconeogenesis. **B.** Similar changes were present in MHet with notable differences

in *gck* and enolases leading to the TCA cycle. **C, D.** While the TCA cycle was affected in MZKO (**C.**), fewer changes were observed in MHet (**D.**).
(TIF)

**S1 Table. Chi square analysis of survival in KOxHet crosses.**
(XLSX)

**S2 Table. Read mapping statistics for RNA-sequencing experiments.**
(XLSX)

**S1 File. Raw counts and DEG lists for RNA sequencing experiments.**
(XLSX)

**S2 File. Gene lists for WGCNA modules.**
(XLSX)

**S3 File. Data from glucose metabolism array.**
(XLSX)

## Acknowledgments

We are incredibly grateful to Kathleen Flaherty of the Rutgers Zebrafish Facility and Rutgers Animal Care staff members for their dedicated work in zebrafish care and the Office of Advanced Research Computing at Rutgers for high performance computing access and maintenance. We also thank our many colleagues and collaborators for insightful discussion on experimental approaches and analyses, including Dr. Clarissa Henry (University of Maine) and Dr. Ronald Hart (Rutgers University).

## Author contributions

**Conceptualization:** Kyle P. Flannery, Namarata Battula, Daniel Burbano, M. Chiara Manzini.

**Data curation:** Kyle P. Flannery, Shorbon Mowla, Namarata Battula, L. Rose Clark, Deze Liu, M. Chiara Manzini.

**Formal analysis:** Kyle P. Flannery, Shorbon Mowla, Namarata Battula, L. Rose Clark, Deze Liu, Daniel Burbano, M. Chiara Manzini.

**Funding acquisition:** M. Chiara Manzini.

**Investigation:** Kyle P. Flannery, Shorbon Mowla, Namarata Battula, L. Rose Clark, Callista D. Oliveira, Lillian M. Simhon, Deze Liu, Cynthia Venkatesan, Brittany F. Karas, Kristin R. Terez, Daniel Burbano, M. Chiara Manzini.

**Methodology:** Kyle P. Flannery, Brittany F. Karas, Kristin R. Terez, Daniel Burbano, M. Chiara Manzini.

**Project administration:** M. Chiara Manzini.

**Supervision:** Daniel Burbano, M. Chiara Manzini.

**Validation:** Kyle P. Flannery, Daniel Burbano, M. Chiara Manzini.

**Visualization:** Kyle P. Flannery, M. Chiara Manzini.

**Writing – original draft:** Kyle P. Flannery, M. Chiara Manzini.

**Writing – review & editing:** Kyle P. Flannery, Shorbon Mowla, Namarata Battula, L. Rose Clark, Callista D. Oliveira, Lillian M. Simhon, Deze Liu, Cynthia Venkatesan, Brittany F. Karas, Kristin R. Terez, Daniel Burbano, M. Chiara Manzini.

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
