## [Decision Letter · Decision Letter 0]

9 Jul 2025

PGENETICS-D-25-00644

Impact of maternal compensation on developmental phenotypes in a zebrafish model of severe congenital muscular dystrophy

PLOS Genetics

Dear Chiara,

Thank you for submitting your manuscript to PLOS Genetics. As you will see, the reviewers are positive about the advance of your study showing that maternal pomgnt2 compensates for zygotic loss of pomgnt2 with the MZ loss reflecting the human condition of severe congenital muscular dystrophy that is not observed with the zygotic loss alone. However, the reviewers make many valuable suggestions to improve the manuscript, including controls, quantification, and clarifications.

Another point from the editors that we would like you to address in a revision is if the results constitute 'maternal genetic compensation'.  It is very clear that the maternally-provided pomgnt2 partially compensates for loss of zygotic pomgnt2 but this is not what is generally considered "genetic compensation". Rather genetic compensation generally refers to upregulation in the expression of another gene, when a related gene is mutated and undergoes nonsense-mediated mRNA decay. Please consider some of the papers from Didier Stainier's lab that address this point.

It is also striking that the maternal pomgnt2 transcript can have an effect apparently so late in larval development.  The editors also think a discussion on this point would also be valuable to readers.

We would be happy to discuss further with you any of these points prior to submission of a revised manuscript.

Therefore, we invite you to submit a revised version of the manuscript that addresses the points raised during the review process.

Please submit your revised manuscript within 90 days Sep 07 2025 11:59PM. If you will need more time than this to complete your revisions, please reply to this message or contact the journal office at plosgenetics@plos.org. Please include the following items when submitting your revised manuscript:

We look forward to receiving your revised manuscript.

Kind regards,

Mary

Mary C. Mullins

Academic Editor

PLOS Genetics

Gregory Cooper

Section Editor

PLOS Genetics

Aimée Dudley

Editor-in-Chief

PLOS Genetics

Anne Goriely

Editor-in-Chief

PLOS Genetics

**Journal Requirements:**

- ® on pages: 31, and 33

- TM on pages: 28, 30, 32, 33, 35, and 36.

3) Thank you for including an Ethics Statement for your study. Please include:

i) The approval number(s), or a statement that approval was granted by the named board(s).

Potential Copyright Issues:

i)  We note that figure 9 is created through BioRender. Please confirm that you hold a Premium account and provide a pdf copy of the CC BY 4.0 Licence as provided by BioRender. For instructions on how to generate a CC BY 4.0 license for your figure, please see the guidelines here: https://help.biorender.com/hc/en-gb/articles/21282341238045-Publishing-in-open-access-resources.

If you are using the free assets from BioRender, we are unable to publish these images as they are licenced under a stricter licence than CC BY 4.0. In this case we ask you to remove the BioRender images and replace them with open source alternatives.

See these open source resources you may use to replace images / clip-art:

- https://bioart.niaid.nih.gov/

- https://bioicons.com/

- https://healthicons.org/

- https://scidraw.io/

- https://reactome.org/icon-lib

- https://www.phylopic.org/images

- https://journals.plos.org/plosbiology/article?id=10.1371/journal.pbio.3002395

6) When completing the data availability statement of the submission form, you indicated that you will make your data available on acceptance. We strongly recommend all authors decide on a data sharing plan before acceptance, as the process can be lengthy and hold up publication timelines. Please note that, though access restrictions are acceptable now, your entire data will need to be made freely accessible if your manuscript is accepted for publication. This policy applies to all data except where public deposition would breach compliance with the protocol approved by your research ethics board. If you are unable to adhere to our open data policy, please kindly revise your statement to explain your reasoning and we will seek the editor's input on an exemption. Please be assured that, once you have provided your new statement, the assessment of your exemption will not hold up the peer review process.

7) Please amend your detailed Financial Disclosure statement. This is published with the article. It must therefore be completed in full sentences and contain the exact wording you wish to be published.

2) If any authors received a salary from any of your funders, please state which authors and which funders..

8) Please ensure that the funders and grant numbers match between the Financial Disclosure field and the Funding Information tab in your submission form. Note that the funders must be provided in the same order in both places as well.

9) Please send a completed 'Competing Interests' statement, including any COIs declared by your co-authors. If you have no competing interests to declare, please state "The authors have declared that no competing interests exist". Otherwise please declare all competing interests beginning with the statement "I have read the journal's policy and the authors of this manuscript have the following competing interests"

**Reviewers' comments:**

Reviewer's Responses to Questions

**Comments to the Authors:**

Reviewer #1: The manuscript by Flannery et al., centers on the generation and characterization of a zebrafish model of severe congenital muscular dystrophy by targeting protein O-mannose N-Acetylglucosaminyltransferase 2 (pomgnt2). POMGNT2 is a maternally provided gene that maintains cell-extracellular matrix interactions through glycosylation and leads to congenital10 muscular dystrophy (CMD) in human patients with pathogenic variants. The authors determined that zygotic knockouts (ZKOs) retain protein function in the first week post fertilization and survive to adulthood, though they develop muscle disease later in life. Interestingly, maternal-zygotic KOs (MZKOs) generated from ZKO females develop early-onset muscle disease, reduced motor function, neuronal axon guidance deficits, and retinal synapse disruptions recapitulating features of the human presentation. The offspring from a ZKO mother, independently of genotype, show distinct expression patterns from animals obtained from heterozygous breedings. The authors determined that these findings implicate genetic compensation and implicate maternal contributions in modeling disease using maternal-zygotic mutants.

This is a very interesting paper with implication for modeling congenital disorders of glycosylation (CDGs) and other human disorders with maternal mRNA contributions. There have been several analyses of previous zebrafish mutant strains that have demonstrated maternal contributions with differential phenotypes, but this is the first for a characterized CDG mutant. Some proper contextualization with other well-documented maternal-zygotic zebrafish mutants with differential phenotypes needs to be discussed. The experiments are logical with mostly appropriate controls and statistical analysis. This manuscript has the potential to be impactful in the CDG and maternal-zygotic impact of generational contributions of key mRNAs/genes.

General Comments:

1. It would be helpful to know if there is genetic compensation from other POMGNT2 pathway members, particularly POMK, POMT1/2, and POMGNT1.

2. The proposed increased energy demand on the ZKO pomgnt2 mutants is an interesting hypothesis, but do the authors actually have data showing increased metabolic demand via ATP measurement comparisons or NAD+ level changes?

3. Have the authors quantitatively evaluated the levels of pomgnt2 mRNA in the maternal KO mutants? Essentially is there any lingering mRNA from the mother that may account for differential phenotypes from the ZKO mutants?

4. Can the authors elaborate a bit more on any neuronal/brain phenotypes, e.g. cobblestone lissencephaly that often occurs in glycosylation disorders.

5. There’s a large amount of literature on differential phenotypes in zebrafish mutant lines due to maternal-zygote effects. Some famous examples are MZzoep, MZspg, mater, and many other early pluripotency genes. The authors should really discuss these cases more and put them in proper contextualization in the discussion section.

6. I’m also a bit surprised that the authors did not compare and contrast more in depth the corresponding authors’ previous work on the pomgnt2/gtdc2 morphant zebrafish (Manzini et al., AJHG, 2012). Given the impact of mRNA contributions, the authors should discuss the phenotypes in a more detailed comparative analysis.

Reviewer #2: In this study, the authors generate and characterize novel zebrafish mutants of pomgnt2, a gene required for the glycosylation of Dystroglycan. Defects in Dystroglycan glycosylation lead to a form of congenital muscular dystrophy, and zebrafish mutants recapitulate both the muscle and CNS pathology seen in human patients and rodent models, making them a good genetic model for the disease. Previous work from this group had found that maternal deposition of mRNA can compensate for loss of pomt1 (another gene required for Dystroglycan glycosylation) in zygotic knockouts. Here they show that this is also the case for pomgnt2: zygotic mutants have a very mild phenotype, whereas mutants generate from KO mothers (maternal-zygotic KOs) show more severe phenotypes that manifests as early as 5 days post-fertilization. They provide a nice side-by-side comparison of these zygotic and maternal-zygotic knockouts, and conduct transcriptomic analysis that identifies a large number of dysregulated genes/pathways in both models that appear before behavioral and anatomical phenotypes are apparent. Overall, this study is well-designed and straightforward, the data are clear, and the findings provide important considerations for how zebrafish models of dystroglycanopathy can be used. There are a couple of issue that should be addressed prior to publication:

• Figure 1: the authors should provide a schematic showing how the crispr frameshift mutations would affect the predicted amino acid sequence

• Figure 1 shows immunostaining for matriglycan (IIH6 antibody) and DAPI. This data nicely shows the loss of glycosylated Dystroglycan in the mutants, but the "variation in myofiber size, gaps between myofibers" (line 135) cannot really be seen here. Can the authors provide a different type of stain (perhaps H&E or laminin) to show the myofibers? Also, they state "increased nuclear staining suggesting fibrosis" (line 136). The increased nuclear staining needs to be quantified.

• Re: Westerns in Fig 1 and S1, the WT samples have significantly less B-DG in all of the westerns provided. Is this a phenotype, or just due to unequal loading? Is there a loading control for these westerns?

• There were a couple points (line 172, 184) where a "trend" towards a difference between genotypes was reported at 10 and 14dpf . These differences are very small and not statistically significant (as opposed to 28dpf). Please remove all mentions of "trends towards significance".

• Interesting that IIH6 is completely gone in the MZKO fish at P7 (Figure 2A), but laminin staining in the muscle still looks fairly normal at this time point (Figure 3E). Could the author comment on this in the discussion? Perhaps there are some clues to upregulated pathways in their transcriptomic analysis that provide clues to how laminin is initially organized in the absence of glycosylated Dystroglycan.

• The quantification in Figure 4 shows that the BTX:SV2 colocalization is more severely disrupted at the MTJ compared to the myofibers. I suggest adding a higher magnification inset for these, as this is somewhat hard to distinguish at the current magnification.

• The authors state: "Horizontal cell nuclei lining the upper border of the inner nuclear layer (INL) also appeared more diffuse and disorganized in MZKOs (Fig. 5A)." (Line 273-275). How do they know these are horizontal cells, as the current staining is only for DAPI? Could they verify this with a horizontal cell marker?

Minor points:

• Supplemental Figure 4 and Figure 8 are intimately related to one another. For ease of readership, it makes sense to name the modules referenced in lines 376-377 by name (ie. blue, green, etc.) so that it is less jarring when the green module is mentioned in the following paragraph if the reader has not downloaded/referenced the supplements.

• Please double check gene name capitalization conventions throughout the introduction and discussion sections. If referring to a human or mouse gene name, capitalize the first letter. If referring to a human gene symbol, capitalize every letter. If referring to a mouse gene symbol, capitalize only the first letter.

• Line 61 refers to Dystroglycan as the extracellular component of the Dystrophin Glycoprotein Complex; it would be more appropriate to refer to Dystroglycan as the transmembrane component of the complex

• The manuscript could benefit from some closer copy editing. There were typos and errors throughout the discussion and methods sections. In the main text the Figure 6 figure legend erroneously refers to panels D-E. They should be C-D.

Some questions regarding the methods:

o. What objectives were used for image acquisition (magnification, NA, air vs immersion)?

o Within the “Weighted gene co-expression network analysis section” it is stated that RStudio was used to perform analysis. RStudio is just the IDE, what version of R was used?

o In the “Quantitative analysis of neuromuscular junctions” section the image acquisition parameters are described including the following sentence: “For each slice, the sample area was scanned four times and averaged with 1 μm intervals.” What does the “averaged with 1um intervals” refer to? Is this pixel binning? Or z-plane averaging? Or does the 1um simply refer to the interval between z-planes, unrelated to the averaging? Please clarify.

o The “Statistical analysis” section mentions the following: “For comparisons between 2 groups of a single measure, an unpaired t-test was used, or alternatively a Mann-Whitney test for nonparametric datasets.” Datasets are not described as nonparametric. Datasets are non-normal. Statistical tests are nonparametric.

o Also in the “Statistical analysis” section - is there a reference for the ROUT method?

**Have all data underlying the figures and results presented in the manuscript been provided?**

Reviewer #1: Yes

Reviewer #2: Yes

PLOS authors have the option to publish the peer review history of their article (what does this mean? ). If published, this will include your full peer review and any attached files.

**Do you want your identity to be public for this peer review?** For information about this choice, including consent withdrawal, please see our Privacy Policy .

Reviewer #1: No

Reviewer #2: No

**Figure resubmission:**
---

## [Decision Letter · Decision Letter 1]

10 Dec 2025

Dear Chiara,

We are pleased to inform you that your manuscript entitled "Impact of maternal compensation on developmental phenotypes in a zebrafish model of severe congenital muscular dystrophy" has been editorially accepted for publication in PLOS Genetics. Congratulations!

With best regards,

Mary

Mary C. Mullins

Academic Editor

PLOS Genetics

Gregory Cooper

Section Editor

PLOS Genetics

Aimée Dudley

Editor-in-Chief

PLOS Genetics

Anne Goriely

Editor-in-Chief

PLOS Genetics

BlueSky: @plos.bsky.social

Comments from the reviewers (if applicable):

Reviewer's Responses to Questions

**Comments to the Authors:**

Reviewer #1: The authors have done a commendable job with regards to addressing my concerns, in particular placing the proper contextualization of this work. No additional concerns.

Reviewer #2: The reviewers have adequately addressed all of my concerns/comments.

**Have all data underlying the figures and results presented in the manuscript been provided?**

Reviewer #1: Yes

Reviewer #2: Yes

PLOS authors have the option to publish the peer review history of their article (what does this mean? ). If published, this will include your full peer review and any attached files.

**Do you want your identity to be public for this peer review?** For information about this choice, including consent withdrawal, please see our Privacy Policy .

Reviewer #1: No

Reviewer #2: No

**Data Deposition**

http://datadryad.org/submit?journalID=pgenetics&manu=PGENETICS-D-25-00644R1

**Press Queries**

---

## [Editor Report · Acceptance letter]

PGENETICS-D-25-00644R1

Impact of maternal compensation on developmental phenotypes in a zebrafish model of severe congenital muscular dystrophy

Dear Dr Manzini,

We are pleased to inform you that your manuscript entitled "Impact of maternal compensation on developmental phenotypes in a zebrafish model of severe congenital muscular dystrophy" has been formally accepted for publication in PLOS Genetics! Your manuscript is now with our production department and you will be notified of the publication date in due course.

With kind regards,

Anita Estes

PLOS Genetics

On behalf of:
